# From Bench to Bedside: Clinical and Biomedical Investigations on Hepatitis C Virus (HCV) Genotypes and Risk Factors for Albuminuria

**DOI:** 10.3390/bioengineering9100509

**Published:** 2022-09-27

**Authors:** Po-Jen Hsiao, Chia-Jen Hsiao, Fu-Ru Tsai, Yen-Lin Hou, Chih-Chien Chiu, Wen-Fang Chiang, Kun-Lin Wu, Yuan-Kuei Li, Chen Lin, Jenq-Shyong Chan, Chi-Wen Chang, Chi-Ming Chu

**Affiliations:** 1Division of Nephrology, Department of Internal Medicine, Armed Forces Taoyuan General Hospital, Taoyuan 325, Taiwan; 2Division of Nephrology, Department of Internal Medicine, Tri-Service General Hospital, National Defense Medical Center, Taipei 114, Taiwan; 3Institute of Cellular and System Medicine, National Health Research Institutes, Miaoli County 350, Taiwan; 4Department of Life Sciences, National Central University, Taoyuan 320, Taiwan; 5School of Medicine, Fu-Jen Catholic University, New Taipei City 242, Taiwan; 6Division of Gastroenterology, Department of Digestive Medicine, New Taipei City Hospital, New Taipei City 241, Taiwan; 7Institute of Molecular and Cellular Biology, National Tsing Hua University, Hsinchu 300, Taiwan; 8Department of Nursing, Armed Forces Taoyuan General Hospital, Taoyuan 325, Taiwan; 9School of Nursing, College of Medicine, Chang Gung University, Taoyuan 333, Taiwan; 10School of Public Health, National Defense Medical Center, Taipei 114, Taiwan; 11Division of Infectious Disease, Department of Internal Medicine, Taoyuan Armed Forces General Hospital, Taoyuan 325, Taiwan; 12Division of Infectious Disease, Department of Internal Medicine, Tri-Service General Hospital, National Defense Medical Center, Taipei 114, Taiwan; 13Division of Colorectal Surgery, Department of Surgery, Taoyuan Armed Forces General Hospital, Taoyuan 325, Taiwan; 14Department of Biomedical Sciences and Engineering, National Central University, Taoyuan 320, Taiwan; 15Division of Pediatric Endocrinology & Genetics, Department of Pediatrics, Chang-Gung Memorial Hospital, Taoyuan 333, Taiwan; 16Graduate Institute of Life Sciences, National Defense Medical Center, Taipei 114, Taiwan; 17Graduate Institute of Medical Sciences, National Defense Medical Center, Taipei 114, Taiwan; 18Department of Public Health, School of Public Health, China Medical University, Taichung 404, Taiwan; 19Department of Public Health, Kaohsiung Medical University, Kaohsiung 807, Taiwan; 20Big Data Research Center, Fu-Jen Catholic University, New Taipei City 242, Taiwan; 21Division of Biostatistics and Medical Informatics, Department of Epidemiology, School of Public Health, National Defense Medical Center, Taipei 114, Taiwan

**Keywords:** hepatitis C virus (HCV) genotype, albuminuria, proteinuria, nephropathies, chronic kidney disease (CKD), National Health and Nutrition Examination Survey (NHANES)

## Abstract

An extrahepatic manifestation of nephropathies can be a feature of the chronic hepatitis C virus (HCV) infection. Albuminuria is a major risk factor for nephropathies and chronic kidney disease (CKD). The correlation between HCV genotypes and albuminuria is still unclear. In this study, investigations have been done for the biomedical tools and methodologies used in the National Health and Nutrition Examination Survey (NHANES) public database. We searched the 2007–2016 NHANES public database to retrieve data regarding the different HCV genotypes and clinical scenarios. This study attempted to investigate the impacts of HCV genetic diversity, associated comorbidities, and racial differences on albuminuria. The urine albumin/creatinine ratio (ACR) was the primary endpoint. Among 40,856 participants, 336 participants with positive and 237 with negative HCV RNA tests were analyzed, excluding 14,454 participants with negative HCV antibodies and 25,828 which were missed. After controlling for sex, race, education level, smoking, diabetes mellitus, hepatitis B, alcohol use, and body mass index (BMI) with a generalized linear equation, HCV genotype 2 was more likely than any other genotype to cause albuminuria based on the urine ACR (*p* < 0.001). The generalized linear equation also demonstrated a significantly higher urine ACR, including hepatitis B (*p* < 0.001), diabetes mellitus (*p* < 0.001), and smoking (*p* = 0.026). In summary, the patients with HCV genotype 2 presented with increased albuminuria in comparison with other HCV genotypes in this 10-year retrospective analysis. HCV infection could be a risk factor of CKD; early diagnosis and appropriate treatment may improve clinical outcomes.

## 1. Introduction

Since its discovery in 1989, remarkable progress has been made in the understanding of the hepatitis C virus (HCV) infection and the development of highly effective direct-acting antivirals (DAAs). However, DAA treatment is still limited for the majority of patients, and approximately 70 million people around the world are still chronically infected with HCV and at increased risk for liver cirrhosis and liver cancer [1,2,3]. An effective vaccine will likely be necessary to achieve global elimination of HCV. At the genetic level, a hallmark of HCV is its extreme diversity. It has been classified into 8 genotypes and 90 subtypes, with an increasing number of additional strains that have yet to be formally classified as subtypes [1,2,3]. A consequence of this diversity is that no vaccine is available to protect against infection. The progress of strategies for HCV infection would also have a major impact on global health. In clinical scenarios, despite the tremendous success of DAAs, there are still challenges that remain to be mentioned. Further epidemiology of the HCV infection has mainly focused on studying the viral genetic diversity. Regardless of host immunology, the genetic diversity of HCV may play a role in the progression of chronic HCV infection and in the extra-hepatic manifestations among these patients [4,5,6]. Previous studies have demonstrated evidence of the relationship between hepatitis C and glomerulonephritis. Glomerulonephritis associated with cryoglobulinemia is the most common nephropathy associated with HCV infection [7,8]. The clinical presentation is proteinuria with microscopic hematuria and a variable degree of renal insufficiency. Chronic kidney disease (CKD) may develop in 10–20% of patients with HCV infection-associated cryoglobulinemic vasculitis. Fabrizio Fabrizi et al., conducted a meta-analysis and recommended that HCV seropositive patients had more frequent proteinuria than patients with negative serology, although the impact of HCV genotypes on proteinuria was not well documented. In addition, no relationship was found between HCV infection and prevalence (or incidence) of decreased estimated glomerular filtration rate (eGFR) [8]. The treatment of HCV-associated glomerulonephritis, especially cryoglobulinemic membranoproliferative glomerulonephritis, encompasses various options, including DAA therapy with or without conventional and novel immunomodulatory agents [1,2,3]. The previous study reported that patients with the HCV infection could be at risk for metabolic syndrome [8]. The Third United States (US) National Health and Nutrition Examination Survey (NHANES) showed that HCV infection is related to albuminuria, especially in populations aged 60 years and older [9]. Taiwanese researchers showed in 1991–1992 that patients with chronic HCV infection are more likely to have higher proteinuria than those without chronic HCV infection. Moreover, a high level of HCV-ribonucleic acid (RNA) (viral load) and genotype 2 have been reported to be a strong predictor of CKD [10]. Contrarily, the REVEAL-HCV study recommended that HCV genotype 1 and high serum HCV RNA levels could be strong predictors of end-stage renal disease (ESRD) [11]. In 2018, a study using data from the National Health and Nutrition Database and the National Vital Statistics System (NVSS) showed that sex and race are significant factors in HCV infection. Increased urinary albumin excretion (albuminuria) is a sensitive marker for renal dysfunction, which has been demonstrated to be a risk factor for progressive kidney injury and can increase further comorbidities and mortality [12]. HCV genotype 1 has been reported to be the most prevalent globally (49.1%), followed by genotype 3 (17.9%), 4 (16.8%), and 2 (11.0%) [1]. In the US, genotype 2 accounts for approximately 13–15% of all HCV infections [1,2]. Different databases and types of methods commonly used by investigators were reviewed to obtain a global picture of biomedical techniques and computational tools used in the HCV human study. Albuminuria is a major risk factor for nephropathies and may cause the progression of CKD in patients with the HCV infection. To date, it remains inconclusive whether there is a difference in kidney outcome between various HCV genotypes. The association between different HCV genotypes and albuminuria is rarely reported, and still needs further examination among different populations with regard to race and ethnicity [10,11]. This study aimed to investigate the clinical characteristics, HCV genotypes, and risk factors for albuminuria in patients with positive HCV RNA compared to those with negative HCV RNA.

## 2. Materials and Methods

### 2.1. Data Source and Subjects

The NHANES, a large public database, is derived from a large national survey. It is an ongoing survey conducted by the Centers for Disease Control and Prevention (CDC) and the National Center for Health Statistics (NCHS) in the U.S.A. [13], with the goal of generating important health statistics. The NHANES program was launched in 1960. The survey was designed for different populations and different health topics. Since 1999, it has been an ongoing survey conducted every other year. It has a large sample size and includes detailed items and multiple dimensions, such as biochemistry results and a questionnaire. In general, items for older participants are more extensive. Some sensitive information may be available upon request, but most information is open to the public and can be downloaded. Participants first complete a health survey at home and then undergo a physical examination at one of four ambulatory centers. The healthcare team is composed of physicians, dentists, nutritionists, hygienists, and laboratory technicians, and the examinations use state-of-the-art, high-tech equipment. The team does not provide healthcare per se, but will provide a copy of the test results to each participant. The staff involved in the survey will also explain the results. All participant information collected during the survey is strictly confidential, and participant privacy is protected by applicable laws. All data of NHANES were collected from survey participants using the questionnaires on health-related topics in participants’ homes, including physical examinations and results of laboratory tests, in a mobile examination center. All of NHANES public data were available on the website of National Center for Health Statistics (available from: https://www.cdc.gov/nchs/nhanes/index.htm accessed on 25 July 2022) and permission for publishing the analysis was not needed. The sample weights in NHANES have been constructed to adjust for non-response, oversampling, and non-coverage. Due to the thoroughness of the research methodology, NHANES data have been widely used over the years to reliably assess many diseases’ prevalence and risk factors. All participants provided written informed consent, and the NHANES protocols were approved by the research ethics review board of the National Center for Health Statistics. The NHANES sample represents the noninstitutionalized civilian U.S.A. population residing in all 50 states and the District of Columbia. In addition, NHANES uses a multifarious, multistage, probability sampling strategy instead of a simple random sample (https://wwwn.cdc.gov/nchs/nhanes/tutorials/module2.aspx accessed on 25 July 2022).

Our study was also conducted according to the declaration of Helsinki. Inclusion criteria selected participants who had the results of their hepatitis C antibody (anti-HCV) or HCV RNA tests. Those with negative HCV RNA tests among the participants with negative anti-HCV were defined as the non-HCV infection group. The participants who had detectable positive HCV RNA tests were defined as the HCV infection group. Participants without any detailed HCV RNA data, including HCV genotypes, or lack of detailed medical information (Section 2.1.1,Section 2.1.2, Section 2.1.3, Section 2.1.4 and Section 2.1.5 ) were excluded from the study. The survey consisted of six items:

#### 2.1.1. Demographics

Participants represent different age groups and have diverse racial backgrounds. The subjects are U.S.A. residents and are categorized by race as Mexican-American, Hispanic-American, non-Hispanic white, non-Hispanic black, other races, and mixed races. Socioeconomic data are also collected, such as education level, income, family size, and military or veteran status.

#### 2.1.2. Dietary Habits

The participants’ usual dietary habits are surveyed. Factors such as a predominantly vegetable versus predominantly meat diet, the ratio of vegetable to meat consumption, the types of food consumed (various grains, roots, and stems), the intake of various foods (calorie intake, water intake [cc]), the intake of various vitamins (such as vitamin A, B, C, D), mealtimes, eating schedule (breakfast, lunch, dinner), and food sources are considered.

#### 2.1.3. Physical Health

Various health indicators are measured, such as general physical examination results, including blood pressure, weight, body mass index (BMI), oral health, bone density (spine, feet, and hands, for some surveys), arthritis assessment, retinal imaging, lung function, and hearing tests.

#### 2.1.4. Biochemistry

The items included in this category are generally available to the public, although the exact items may vary from year to year. The collected data include urine albumin and creatinine, serum lead, cholesterol, fluorides, folic acid, hepatitis A, hepatitis B, hepatitis C, herpes simplex virus, human immunodeficiency virus (HIV), human papillomavirus, insulin, routine tests (blood test), and volatile organic compounds. The test of urine albumin/creatinine ratio (ACR) was determined by quantitative method.

#### 2.1.5. Questionnaire

The items collected information regarding alcohol use, hearing test results, blood pressure and cholesterol, cardiovascular health, consumer behaviour, current health status, dermatology, diabetes, dietary behaviour and nutrition, disability, drug use, health insurance, hepatitis, kidney disease, mental health, reproductive health, sexual behaviour, smoking, sleep disorders, and weight history.

#### 2.1.6. Restricted Access and Patient Data Collection

For the protection the privacy of the participants, some participant data are only available upon request to download for analysis purposes. Otherwise, users may view the distribution of samples, including geographic data, alcohol use (for young participants), reproductive health (for pregnant women), sexual behaviour (for young participants), and urine trichomonas (for young participants). According to Thomas Frieden, director of the CDC, “The NHANES enters the community to obtain health information across the country, thereby becoming a national ‘health check-up’. The survey is a unique health information resource. Without it, we would lack important understanding of major health conditions.” Public health officials, legislators, and physicians use the information collected by NHANES to develop health policies, guide and design public health programs and services, and promote health and hygiene knowledge across the country. In addition, NHANES data are used to establish a standardized growth chart for pediatricians across the country to use to track the growth and development of children. In recent years, the NHANES has focused on the health status of elderly adults, Asian Americans, and African Americans.

In the present study, we collected 10-year data from five surveys: 2007–2008, 2009–2010, 2011–2012, 2013–2014, and 2015–2016, with approximately 5000 samples per year. We used the probability sampling method to sample representative residents from 15 counties across the U.S.A. The data represented all age groups in the U.S.A. and excluded nursing home residents, members of the military, and expatriates [14]. A stratified multistage probability sampling design was used to randomly select households with a probability proportional to size (PPS). Next, we randomly selected participants from each subgroup, with an average of 1.6 participants from each household [15]. As noted in the study title, we included samples with or without hepatitis C (based on hepatitis C testing from section IV, biochemistry). Any subgroup with a sample size of fewer than 5 was excluded from subsequent analyses. The study flow chart is described below:

### 2.2. Study Tools

Basic information (age, sex, race, and education level), physical health (BMI), biochemistry (HBV, HCV, HIV, and urine ACR), and the questionnaire (alcohol use, smoking, diabetes mellitus, hypertension, and drug use) were analyzed in this study.

### 2.3. Operational Definition of Study Variables

#### 2.3.1. Data Processing and Variable Selection

In this study, we collected relevant data from the 2007–2016 NHANES database, with the serial sequence number (SEQN) as the string field. Each category consisted of multiple items, and each variable in the relevant items was downloaded as a separate. xpt file, which was then converted to a SAS file for SPSS v24.0. The six categories for each year were merged, and then the 10-year data were merged. Next, the hepatitis C field in section III, biochemistry, was selected for data analysis after missing values were excluded.

#### 2.3.2. Statistical Analysis

Required data from the target database were selected. After the data were reviewed for completeness, IBM Statistics SPSS v24.0 was used for data classification, variable conversion, redefining variables, observation value selection, and statistical analysis, including descriptive analysis and inferential analysis, according to the specified purpose of the study. A value of *p* < 0.05 (two-tailed) was considered statistically significant. A multivariable generalized linear model was performed in the descriptive analysis [16]. We calculated the frequency and percentage of factors including sex, race, HCV genotype, education level, smoking, alcohol use, diabetes mellitus, hypertension, drug use, HIV, HBV, and BMI. Furthermore, we analyzed the means and standard deviation of continuous variables, age and urine ACR. In the inferential analysis, we used the Chi-square test and the Kruskal-Wallis test to analyze the relationship between the HCV genotypes and urine ACR.

## 3. Results

### 3.1. Clinical Characteristics

The study flow chart is shown in Figure 1. We used the 2007–2016 database of U.S.A. residents of various races. There were more Caucasian participants than other races, followed in number by participants of African descent. A total of 40,856 participants were initially included in this study; participants with missing hepatitis C values or lack of detailed medical records were excluded. A total of 14,454 screening participants had negative hepatitis C antibody tests (anti-HCV) and the remaining subjects (n = 237) were categorized into the non-HCV infection group based on negative HCV RNA tests. Participants with positive HCV RNA tests (n = 336) were evaluated as the HCV infection group. Finally, the total subjects (n = 573) were divided into two groups based on positive or negative HCV RNA tests in order to analyze the distribution for each variable. The incidence of hepatitis C was 1.4%. The baseline socio-demographic variables and distribution of HCV genotypes is shown in Table 1.

#### 3.1.1. Gender

Among male participants, genotype 1a was identified in 141 men (69.1%), genotype 1b was identified in 36 men (57.1%), genotype 2 was identified in 20 men (66.7%), and genotype 3 was identified in 14 men (51.9%); among female participants, genotype 1a was identified in 63 women (30.9%), genotype 1b was identified in 27 women (42.9%), genotype 2 was identified in 10 women (33.3%), and genotype 3 was identified in 13 women (48.1%).

#### 3.1.2. Age

Among the four age groups, no participants in the <20-year age group had hepatitis C. For the 20- to 39-year age group, genotype 1a was identified in 23 participants (11.3%), genotype 1b was identified in 1 participant (1.6%), genotype 2 was identified in 4 participants (13.3%), and genotype 3 was identified in 3 participants (11.1%). For the 40- to 59-year age group, genotype 1a was identified in 116 participants (56.9%), genotype 1b was identified in 33 participants (52.4%), genotype 2 was identified in 15 participants (50.0%), and genotype 3 was identified in 19 participants (70.4%). For the 60 years and older age group, genotype 1a was identified in 65 participants (31.9%), genotype 1b was identified in 29 participants (46.0%), genotype 2 was identified in 11 participants (36.7%), and genotype 3 was identified in 5 participants (18.5%)

#### 3.1.3. Race

The survey included five categories, including Mexican-American, Hispanic, non-Hispanic white, non-Hispanic black, non-Hispanic Asian, and other. Among Mexican-American participants, genotype 1a was identified in 17 participants (9.0%), genotype 1b was identified in 7 participants (11.3%), genotype 2 was identified in 3 participants (10.0%), and genotype 3 was identified in 6 participants (23.1%). Among Hispanic participants, genotype 1a was identified in 18 participants (9.0%), genotype 1b was identified in 3 participants (4.8%), genotype 2 was identified in 5 participants (16.7%), and genotype 3 was identified in 4 participants (15.4%). Among non-Hispanic white participants, genotype 1a was identified in 75 participants (37.7%), genotype 1b was identified in 13 participants (21.0%), genotype 2 was identified in 19 participants (63.3%), and genotype 3 was identified in 11 participants (42.3%). Among non-Hispanic black participants, genotype 1a was identified in 89 participants (44.7%), genotype 1b was identified in 38 participants (61.3%), genotype 2 was identified in 2 participants (6.7%), and genotype 3 was identified in 2 participants (7.7%). Among other races, genotype 1a was identified in 5 participants (2.5%), genotype 1b was identified in 2 participants (3.2%), genotype 2 was identified in 1 participant (3.3%), and genotype 3 was identified in 4 participants (14.8%).

#### 3.1.4. Education Level

The survey included three categories: high school, college or equivalent, and post-graduate. Among the participants with a high school education, genotype 1a was identified in 139 participants (68.1%), genotype 1b was identified in 38 participants (60.3%), genotype 2 was identified in 22 participants (75.9%), and genotype 3 was identified in 21 participants (80.8%). Among the participants with a college or equivalent education, genotype 1a was identified in 57 participants (27.9%), genotype 1b was identified in 19 participants (30.2%), genotype 2 was identified in 5 participants (17.2%), and genotype 3 was identified in 4 participants (15.4%). Among the participants with a post-graduate education, genotype 1a was identified in 8 participants (3.9%), genotype 1b was identified in 6 participants (9.5%), genotype 2 was identified in 2 participants (6.9%), and genotype 3 was identified in 1 participant (3.8%).

#### 3.1.5. Smoking

Among smokers, genotype 1a was identified in 123 participants (73.2%), genotype 1b was identified in 29 participants (60.4%), genotype 2 was identified in 19 participants (70.4%), and genotype 3 was identified in 16 participants (69.6%). Among non-smokers, genotype 1a was identified in 45 participants (26.8%), genotype 1b was identified in 19 participants (39.6%), genotype 2 was identified in 8 participants (29.6%), and genotype 3 was identified in 7 participants (30.4%).

#### 3.1.6. Drug Use

The distribution of drug use was not analyzed due to missing data for non-users and the small sample size.

#### 3.1.7. Diabetes Mellitus

Among diabetic participants, genotype 1a was identified in 29 participants (14.2%), genotype 1b was identified in 13 participants (20.6%), genotype 2 was identified in 1 participant (3.3%), and genotype 3 was identified in 6 participants (22.2%). Among non-diabetic participants, genotype 1a was identified in 175 participants (85.8%), genotype 1b was identified in 50 participants (79.4%), genotype 2 was identified in 29 participants (96.7%), and genotype 3 was identified in 21 participants (77.8%).

#### 3.1.8. Hypertension

Among hypertensive participants, genotype 1a was identified in 96 participants (88.9%), genotype 1b was identified in 33 participants (84.6%), genotype 2 was identified in 7 participants (50.0%), and genotype 3 was identified in 9 participants (90.0%). Among non-hypertensive participants, genotype 1a was identified in 12 participants (11.1%), genotype 1b was identified in 6 participants (15.4%), genotype 2 was identified in 7 participants (50.0), and genotype 3 was identified in 1 participant (10.0%).

#### 3.1.9. Hepatitis B

Among participants with hepatitis B, genotype 1a was identified in 2 participants (1.1%), genotype 1b was identified in 0 participants, genotype 2 was identified in 0 participants, and genotype 3 was identified in 1 participant (4.3%). Among participants without hepatitis B, genotype 1a was identified in 184 participants (84.6%), genotype 1b was identified in 59 participants (20.3%), genotype 2 was identified in 25 participants (8.6%), and genotype 3 was identified in 22 participants (7.6%).

#### 3.1.10. HIV

Among participants with HIV, genotype 1a was identified in 4 participants (3.3%), genotype 1b was identified in 1 participant (3.7%), genotype 2 was identified in 0 participants, and genotype 3 was identified in 0 participants. Among participants without HIV, genotype 1a was identified in 119 participants (96.7%), genotype 1b was identified in 26 participants (96.3%), genotype 2 was identified in 17 participants (100%), and genotype 3 was identified in 18 participants (100%).

#### 3.1.11. Alcohol Use

Among alcohol users, genotype 1a was identified in 176 participants (91.7%), genotype 1b was identified in 50 participants (83.3%), genotype 2 was identified in 22 participants (78.6%), and genotype 3 was identified in 17 participants (73.9%). Among non-drinkers, genotype 1a was identified in 16 participants (8.3%), genotype 1b was identified in 10 participants (16.7%), genotype 2 was identified in 6 participants (21.4%), and genotype 3 was identified in 6 participants (26.1%).

#### 3.1.12. BMI

The survey included three categories: underweight, normal weight, and overweight. In the underweight group, genotype 1a was identified in 4 participants (2.0%), genotype 1b was identified in 1 participant (1.6%), genotype 2 was identified in 0 participants, and genotype 3 was identified in 0 participants. In the normal weight group, genotype 1a was identified in 47 participants (23.4%), genotype 1b was identified in 12 participants (19.4%), genotype 2 was identified in 9 participants (31.0%), and genotype 3 was identified in 7 participants (25.9%). In the overweight group, genotype 1a was identified in 150 participants (74.6%), genotype 1b was identified in 49 participants (79%), genotype 2 was identified in 20 participants (69.0%), and genotype 3 was identified in 20 participants (74.1%).

#### 3.1.13. Liver Function Tests and Lipid Profiles

Participants with positive HCV RNA tests had significant higher liver function tests (AST, ALT) and lower triglyceride levels compared to those with negative HCV RNA tests. However, there were no obvious significances between the different HCV genotypes (Appendix A).

#### 3.1.14. Urine ACR

The survey included three groups: <30 mg/g (normoalbuminuria), 30–300 mg/g (microalbuminuria), and >300 mg/g (macroalbuminuria). Genotype 1a was identified in 158 participants (79.8%) in the <30 mg/g group, 32 participants (16.2%) in the 30–300 mg/g group, and 8 participants (4%) in the >300 mg/g group; genotype 1b was identified in 45 participants (72.6%) in the <30 mg/g group, 15 participants (24.2%) in the 30–300 mg/g group, and 2 participants (3.2%) in the ≥300 mg/g group; genotype 2 was identified in 23 participants (82.1%) in the <30 mg/g group, 4 participants (14.3%) in the 30–300 mg/g group, and 1 participant (3.6%) in the >300 mg/g group; genotype 3 was identified in 24 participants (88.9%) in the <30 mg/g group, 3 participants (11.1%) in the 30–300 mg/g group, and 0 participants in the >300 mg/g group.

### 3.2. Distribution of HCV Genotypes

The dominant genotype was type 1 (1a: 60.9%, 1b: 18.8%). The proportions of type 2 and type 3 were similar (9.0% versus 8.1%) (Table 2).

### 3.3. HCV Genotypes and Urine ACR

In order to analyze the relationship between HCV genotypes and urine ACR, the values of these biochemical indicators are expressed as mean ± standard deviation. For participants with positive HCV RNA (n = 336), the mean urine ACR was 129.3722 mg/g. Genotype 1a was identified in 200 participants (mean ACR: 124.9069 mg/g); genotype 1b was identified in 62 participants (mean ACR: 39.7754 mg/g); genotype 2 was identified in 29 participants (mean ACR: 501.6887 mg/g); and genotype 3 was identified in 27 participants (mean ACR: 13.0804 mg/g). For participants with negative HCV RNA (n = 237), the mean urine ACR was 38.0117 mg/g.

### 3.4. Generalized Linear Equation of the Relationship between HCV Genotype and Urine ACR

The results of the normality test of the linear model invalidated the hypothesis of normality. Therefore, the generalized linear model was used to analyze which HCV genotypes might predict the relationship among urine ACR. Hypertension, drug use, and HIV were excluded from analysis because of the high number of missing values and subsequent small sample sizes.

With the negative HCV RNA group as the control group, the B value was 100.054 for genotype 1a, −44.124 for genotype 1b, 257.495 for genotype 2, and −194.592 for genotype 3. The albumin level was highest for genotype 2 (B = 257.495), although the *p*-values did not reach statistical significance.

For education level, after controlling for other variables, the B value was 260.312 for participants with a high school education, which was higher than that of participants with a college or equivalent education (227.930, *p* = 0.05). For diabetes, after controlling for other variables, the B value was 494.687 for diabetic participants, which was significantly higher than that of non-diabetic participants (*p* < 0.001). For hepatitis B, after controlling for other variables, the B value was 3124.922 for participants with hepatitis B, which was significantly higher than that of participants without hepatitis B (*p* < 0.001).

Using the generalized linear model (Table 3), the independent variables were HCV genotype, sex, age, race, education level, smoking, diabetes, hepatitis B, alcohol use, and BMI; the dependent variable was urine ACR. After controlling for sex, age, race, education level, smoking, diabetes, hepatitis B, alcohol use, and BMI, the omnibus test of the coefficient showed *p* < 0.001, thereby invalidating the null hypothesis and indicating that the model was significant and had predictive power. With the negative HCV RNA group as the control group, the B value was 66.698 for genotype 1a, −29.669 for genotype 1b, 635.457 for genotype 2, and −157.011 for genotype 3. ACR was highest for genotype 2 (B = 635.457; *p* < 0.001). The *p*-values of the other three groups did not reach statistical significance. After controlling for other variables, HCV genotype 2 was significantly related to urine ACR.

For diabetes mellitus, after controlling for other variables, the B value was 592.153 for diabetic participants, which was significantly higher than that of non-diabetic participants (*p* < 0.001). For hepatitis B, after controlling for other variables, the B value was 1894.796 for participants with hepatitis B, which was significantly higher than that of participants without hepatitis B (*p* < 0.001).

## 4. Discussion

Several histologic types of glomerular diseases are associated with HCV infection, the most frequent being type 1 membranoproliferative glomerulonephritis, usually in the context of Type 2 mixed cryoglobulinemia. The pathogenesis can be attributed to glomerular deposition of cryoglobulins or noncryoglobulin-immune complexes. Cryoglobulins classically comprised immunoglobulin Mκ with rheumatoid factor activity. Patients may present with microscopic hematuria, proteinuria, hypertension, and acute nephrotic and/or nephritic syndrome [7,8]. Proteinuria refers to increased excretion of all proteins through the urine, while albuminuria refers to increased urinary excretion of albumin, which is the predominant urinary protein [17,18,19]. Albuminuria is a sensitive marker for renal dysfunction and has diagnostic, therapeutic, and prognostic significance [20,21,22,23]. There is a linear correlation between albuminuria and all-cause mortality, as well as cardiovascular mortality independent of estimated GFR (eGFR) [24]. Current guidelines on CKD from the Kidney Disease Improving Global Outcomes (KDIGO) recommend measuring albuminuria to quantify proteinuria, since albumin is the predominant protein in the vast majority of proteinuric kidney diseases [25]. To date, the large epidemiologic human study for the investigation of the association between HCV, cryoglobulinemia, and kidney diseases is still lacking. To our knowledge, this is the first study to investigate the association between urine ACR and different HCV genotypes. In addition, our study is the first to show that HCV genotype 2 may be a strong predictor of the presence of albuminuria.

### 4.1. HCV-Associated Nephropathies

Previous studies have demonstrated further evidence of the relationship between hepatitis C and progression to kidney failure [8,9,10,11]. There are several mechanisms of HCV-induced kidney damage (Table 4). The most well documented HCV-related glomerulopathy was type I membranoproliferative glomerulonephritis associated with type II mixed cryoglobulinemia [7,8]. Nevertheless, only selected cryoglobulinemic patients developed glomerular injury, while the majority had nonspecific clinical presentations. The evidence from HCV RNA and associated proteins in endothelial, mesangial, and tubular cells of the kidney tissue may indicate a direct cytopathic effect of HCV invasion. HCV itself also can enter the infected cells and replicate in B lymphocytes. B cell activation during acute and chronic HCV infection then leads to polyclonal activation and expansion of B cells producing monoclonal IgM rheumatoid factors (RF). Monoclonal IgMs are known to display RF activity, favoring the formation of cold-precipitable immune complexes. These are formed by the monoclonal IgM itself, HCV, and anti-HCV polyclonal IgG antibodies, which are responsible for glomerulonephritis. The classical complement pathway is activated when the cold-precipitable immune complexes are deposited in the mesangial and subendothelial space. When complement component C4 is cleaved by cold-precipitable immune complexes, it forms two fragments, C4a and C4b. This then leads to the downstream formation of C3 and C5 convertases. C3a and C5a activate the complement cascade via C3aR and C5aR, which amplify the inflammatory response by activating neutrophils. The interaction of the membrane attack complex (MAC) with C3b and C5b enhances tissue factor (TF) expression in endothelial cells (ECs) of the kidney and can further promote proinflammatory EC activation, including changes in reactive oxygen species (ROS), chemokines, and nitric oxide (NO) (Figure 2). Moreover, the hybridization signals of HCV RNA in renal biopsies were detected in tubular and capillary endothelial cells [7,8,9]. This evidence proposes that HCV may cause glomerular as well as tubulointerstitial injury of the kidney through cryoglobulins, the HCV antibody immune complex, or a direct cytopathic effect. Consequently, HCV infection, especially in active viral reaction, may lead to kidney injury and deterioration of kidney function. Patients with a chronic HCV infection can also have an increased risk of insulin resistance, which may develop during the inflammatory process and aggravate kidney injury [26,27]. A previous human study also recommended that a high HCV viral load and HCV genotype 2 could be potential predictors of CKD in Taiwan [10]. Our study results also demonstrated that HCV genotype 2 was strongly correlated with urine ACR. Diabetes mellitus and hepatitis B could also be factors which might affect the degree of albuminuria in patients with HCV infections. We suggest that a proteinuria or albuminuria screening test is indicated for patients with chronic HCV infections. The NHANES III showed that hepatitis C was related to albuminuria in populations aged 40 years and above. For hepatitis C, seropositivity increases the risk of proteinuria 1.47-fold (CI: 1.12–1.94, *p* = 0.006) [19]. Few studies have investigated the relationship between HCV genotypes and albuminuria. From recent research, Chen YC et al. performed a population-based cross-sectional study using the NHANES data from 1999–2018 in order to investigate the association between the status of HCV infection and the risk of kidney disease. Prevalent eGFR < 60 mL/min/1.73 m^2^ or urinary albumin/creatinine ratio ≥ 30 mg/g was defined as kidney disease in this study. They concluded that genotype 1, in both resolved and chronic HCV infections, was associated with higher risk of kidney disease [28]. In our study, the results show that genotype 2 may have greater risk of albuminuria, instead of focusing on eGFR decline in patients with an HCV infection. These different study results may be due to the original retrospective designs, different endpoints, selection bias, or unmeasured confounding factors such as the inclusion and exclusion of criteria. In our study, after adjustment of the confounding factors, including sex, race, education level, smoking, diabetes mellitus, hepatitis B, alcohol use, and BMI, HCV genotype 2 was confirmed to be strongly associated with ACR, which may have important clinical significance.

### 4.2. Management of HCV-Associated Nephropathies

A better understanding of the pathophysiology of HCV-associated kidney diseases has gradually opened the door for more targeted, hypothesis-driven approaches: (a) antiviral therapy to avoid the formation of cryoglobulins, immune complex deposits, and/or direct HCV injury to the kidney; (b) B-cell depletion, aimed at decreasing cryoglobulin production; and (c) immunosuppressive therapy targeting the glomerular associated inflammation [29,30,31]. Recent data from relatively small studies demonstrated promise of the novel antiviral therapies in HCV-associated glomerulopathies. Antiviral treatments combined with B-cell depleting agents may improve clinical outcomes of these affected patients. It has been around 5–6 years since the first approved use of DAAs in the U.S.A., and some data have been shown that they have an effective control of HCV-associated kidney diseases. Most studies have shown a stabilization of or mild rise in the eGFR rather than a return to normal kidney function [32,33,34,35,36,37]. Sise ME et al. performed a retrospective observational cohort study, which demonstrated that albuminuria improved significantly in patients without diabetes mellitus, but not in those with diabetes mellitus. Predictors of eGFR improvement included having CKD at baseline and being non-diabetic. They concluded that DAA treatment for HCV infection may slow down the progression of CKD [38]. Recently, Liu CH et al. conducted a prospective cohort study among 1987 patients with eGFR ≥15 mL/min/1.73 m^2^ receiving interferon or DAA treatment. They also recommended that the kidney function and risk of ESRD were significantly improved in patients who completed sustained virologic response with anti-HCV therapy [39]. In summary, CKD has an unavoidably progressive natural course, and studies that fail to prove a favorable effect in the post-treatment period should be interpreted by comparing them with studies that analyzed eGFR slopes before and after DAA treatment. The results of these studies recommend the benefits of DAA treatment in slowing down the renal function decline [37,38,39].

### 4.3. Strengths and Limitations

NHANES is the only representative database in the U.S.A. with a large sample size and diverse survey items. However, the database has some limitations. As a cross-sectional study, NHANES only estimates the incidence of HCV infection based on the current survey. It is unable to determine the temporal sequence of exposure to risk factors and HCV infection. In people with anti-HCV (-), the HCV RNA (+) rate is quite low, making the false positive rate of actual HCV carriers extremely low, although some patients with an acute HCV infection may have anti-HCV seroreversion from anti-HCV (+) to anti-HCV (-), and are expected to have a very limited time frame of HCV exposure. Therefore, if we try to evaluate the effect of chronic HCV infection, all participants with anti-HCV (-) antibodies should be included in the control group. If we try to assess the effect of HCV RNA (+), then those with anti-HCV (-) antibodies and HCV RNA (-) should be in the control group, not only the HCV RNA (-) group. This may affect the study results, since only 336 patients with viremia were included in the study. In the present study, we collected 10-year data from five surveys between 2007 and 2016, with approximately 5000 samples per year. The U.S. Food and Drug Administration (FDA) approved most DAA treatment in 2014–2015; therefore, we did not evaluate the DAA data in this study. Moreover, results derived from the NHANES database are applicable in the U.S.A., but not in Taiwan. This is due to variations in genotype distribution and race (in the West versus in Southeast Asia). Furthermore, NHANES participants are taken from the general population, while certain high-risk and high-prevalence populations, such as homeless people and inmates, are excluded. As a result, some HCV patients are excluded from the survey. Moreover, drug use is not socially acceptable; therefore, some individuals may not want to admit to this behaviour, which may result in its underestimation. A timed urine sample allowing estimation of the albumin excretion rate (AER) is the gold standard for quantifying albuminuria. In clinical scenarios, the collection of urine for AER may be impractical. Previous studies have shown a strong correlation between AER and the urine ACR, and consequently, urine ACR is considered reliable for quantifying albuminuria [15,17,18,19,20]. In addition, 24 h urine collection is frequently limited by patients’ adherence, and may have some potential mistakes in clinical practice. During statistical analysis, any outliers cause right skewness due to the small sample size in general as well as the small sample size of certain genotypes. Therefore, large studies are needed to obtain a normal distribution and more accurate analyses. Medication may affect the correlation analysis. However, the NHANES database has no information on HCV, HBV, or HIV medication. Therefore, we are unable to further clarify the relationship between HCV genotypes and HCV in the presence of these interfering factors. HCV single-strand RNA (~9.6 kb) codes for a single large polyprotein, undergoing proteolytic processing to produce three structural (core, E1, and E2) and seven nonstructural proteins (p7, NS2, NS3, NS4A, NS4B, NS5A, and NS5B). The HCV core antigen is a serological marker of infection. To date, at least 8 genotypes and 90 subtypes of HCV have been characterized by clinical manifestations and different geographic distributions. Genotype 1 is prevalent in Japan, the U.S.A., and Europe [1,2,30,31,40]. Therefore, the patient numbers with HCV genotype 2 were relatively low in this study. Further, the application of bioinformatics software may have been an effective tool to investigate the potential biological effects of HCV genomes, including different core domains on albuminuria. Finally, different ethnic backgrounds may have variation due to human genetic or environmental effects.

## 5. Conclusions

Chronic HCV infection remains an important public health issue, which is associated with extrahepatic manifestations including cryoglobulinemia, lymphoproliferative disorders, and renal diseases. The most common HCV-associated nephropathy is cryoglobulinemic glomerulopathy, but HCV infection can cause other types of glomerulopathies. This biomedical analysis of the NHANES database, including a representative sample of the U.S.A. population, demonstrated that patients with HCV genotype 2 significantly presented with increased albuminuria. Comorbidities including hepatitis B, diabetes mellitus, and smoking have impacts on albuminuria. Urinalysis should be part of routine checks, especially in these patients. Early recognition and treatment may provide more favorable outcomes for the patients. Further animal or human studies may be required to investigate the alterations in albuminuria and associated kidney diseases in patients with chronic HCV infections.

## Figures and Tables

**Figure 1 bioengineering-09-00509-f001:**
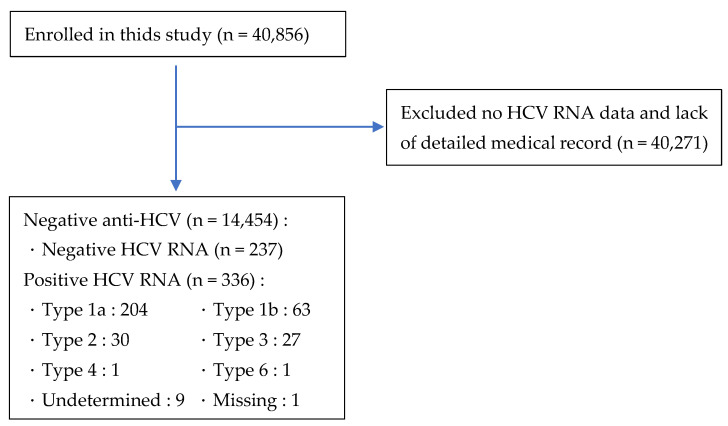
Study flow chart.

**Figure 2 bioengineering-09-00509-f002:**
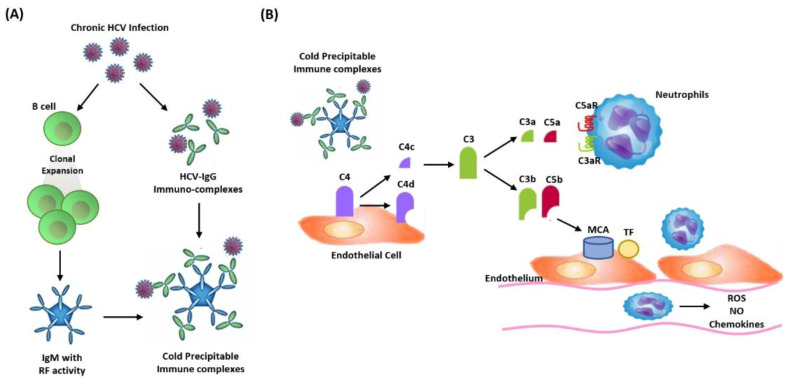
Molecular mechanisms of HCV-associated cryoglobulinemic glomerulonephritis. (**A**) B cell activation during acute and chronic HCV infection, which then leads to polyclonal activation and expansion of B cells producing monoclonal IgM rheumatoid factor (RF). Monoclonal IgMs that are known to display RF activity, favoring the formation of cold-precipitable immune complexes formed by the monoclonal IgM itself, HCV, and anti-HCV polyclonal IgG antibodies, which are responsible for glomerulonephritis. (**B**) The classical complement pathway is activated when the cold-precipitable immune complexes are deposited in the mesangial and subendothelial space in the kidney. When complement component C4 is cleaved by cold-precipitable immune complexes, it forms two fragments, C4a and C4b. This then results in the downstream formation of C3 and C5 convertases. C3a and C5a activate the complement cascade via C3aR and C5aR, which amplifies the inflammatory response by activating neutrophils. The interaction of the membrane attack complex (MAC) with C3b and C5b enhances tissue factor (TF) expression in endothelial cells (ECs) of the kidney and can further promote proinflammatory EC activation.

**Table 1 bioengineering-09-00509-t001:** Socio-demographic variables and distribution of HCV genotypes.

All = 573	Positive (%)	Negative (%)	*p*-Value	Genotype 1a	Genotype 1b	Genotype 2	Genotype 3	*p*-Value
	n = 336	n = 237		n = 204	n = 63	n = 30	n = 27	
Gender								
Male	222 (66.1)	131 (55.3)	<0.009 **	141 (69.1)	36 (57.1)	20 (66.7)	14 (51.9)	0.15
Female	114 (33.9)	106 (44.7)		63 (30.9)	27 (42.9)	10 (33.3)	13 (48.1)	
Age (years)								
<20	0	19 (8)	<0.001 ***	-	-	-	-	0.072
20–39	32 (9.5)	46 (19.4)		23 (11.3)	1 (1.6)	4 (13.3)	3 (11.1)	
40–59	191 (56.8)	93 (39.2)		116 (56.9)	33 (52.4)	15 (50.0)	19 (70.4)	
≥60	113 (33.6)	79 (33.3)		65 (31.9)	29 (46.0)	11 (36.7)	5 (18.5)	
Race								
Mexican-American	34 (10.1)	33 (13.9)	<0.001 ***	17 (8.5)	7 (11.3)	3 (10.0)	6 (23.1)	<0.001 ***
Hispanic	31 (9.2)	24 (10.1)		18 (9.0)	3 (4.8)	5 (16.7)	4 (15.4)	
Non-Hispanic white	124 (36.9)	116 (48.9)		75 (37.7)	13 (21.0)	19 (63.3)	11 (42.3)	
Non-Hispanic black	133 (39.6)	42 (17.7)		89 (44.7)	38 (61.3)	2 (6.7)	2 (7.7)	
Other	14 (4.2)	22 (9.3)		5 (2.5)	2 (3.2)	1 (3.3)	4 (14.8)	
Education level								
High school	228 (68.3)	117 (53.7)	<0.001 **	139 (68.1)	38 (60.3)	22 (75.9)	21 (80.8)	0.306
College or equivalent	88 (26.3)	68 (31.2)		57 (27.9)	19 (30.2)	5 (17.2)	4 (15.4)	
Post-graduate	18 (5.4)	33 (15.1)		8 (3.9)	6 (9.5)	2 (6.9)	1 (3.8)	
Smoking								
Yes	116 (49.4%)	72 (30.4%)	<0.001 ***	123 (73.2)	29 (60.4)	19 (70.4)	16 (69.6)	0.402
No	170(50.6%)	165 (69.6%)		45 (26.8)	19 (39.6)	8 (29.6)	7 (30.4)	
Diabetes mellitus								
Yes	51 (15.2%)	30 (12.7%)	0.386	29 (14.2)	13 (20.6)	1 (3.3)	6 (22.2)	0.115
No	284 (84.8%)	207 (87.3%)		175 (85.8)	50 (79.4)	29 (96.7)	21 (77.8)	
Hypertension								
Yes	151 (85.3%)	74 (76.3%)	0.062	96 (88.9)	33 (84.6)	7 (50.0)	9 (90.0)	0.002 *
No	26 (14.7%)	23 (23.7%)		12 (11.1)	6(15.4)	7 (50.0)	1 (10.0)	
Hepatitis B								
Yes	4 (1.3%)	2 (1.1%)	0.8	2 (1.1)	0	0	1 (4.3)	0.336
No	300 (98.7%)	187 (98.9%)		184 (63.4)	59 (20.3)	25 (8.6)	22 (7.6)	
HIV								
Yes	5 (2.6%)	4 (2.9%)	0.865	4 (3.3)	1 (3.7)	0	0	0.749
No	188 (97.4%)	134 (97.1%)		119 (96.7)	26 (96.3)	17 (100)	18 (100)	
Alcohol use								
Yes	272 (86.9)	162 (77.9)	<0.007 ** 0.729	176 (91.7)	50 (83.3)	22 (78.6)	17 (73.9)	0.019 * 0.911
Week	97 (49.2)	55 (46.2)		63 (49.2)	19 (54.3)	6 (35.3)	6 (50.0)	
Month	36 (19.8)	22 (18.5)		26 (20.3)	6 (17.1)	5 (29.4)	2 (16.7)	
Year	61 (31)	42 (35.3)		39 (30.5)	10 (28.6)	6 (35.3)	4 (33.3)	
No	41 (13.1)	46 (22.1)		16 (8.3)	10 (16.7)	6 (21.4)	6 (26.1)	
BMI								
Underweight	5 (1.5)	12 (5.1)	<0.039 *	4 (2.0)	1 (1.6)	0	0	0.856
Normal weight	81 (24.5)	60 (25.6)		47 (23.4)	12 (19.4)	9 (31.0)	7 (25.9)	
Overweight	245 (74.0)	162 (69.2)		150 (74.6)	49 (79)	20 (69.0)	20 (74.1)	
Urine ACR								
<30 mg/g	260 (79.5)	192 (82.1)	0.723	158 (79.8)	45 (72.6)	23 (82.1)	24 (88.9)	0.617
30–300 mg/g	56 (17.1)	36 (15.4)		32 (16.2)	15 (24.2)	4 (14.3)	3 (11.1)	
>300 mg/g	11 (3.4)	6 (2.6)		8 (4.0)	2 (3.2)	1 (3.6)	0	

* *p*-Value < 0.05, ** *p*-Value < 0.01, *** *p*-Value < 0.001. Percentages (%) are listed in a column to make it easy to see which category has the highest percentage. Abbreviations: ACR: albumin/creatinine ratio; BMI: body mass index (underweight: <18.5, normal weight: 18.5–24.9, overweight: ≥25); HIV: human immunodeficiency virus.

**Table 2 bioengineering-09-00509-t002:** Distribution of HCV genotypes.

All = 335	N (%)
Genotype	
Genotype 1a	204 (60.9)
Genotype 1b	63 18.8)
Genotype 2	30 (9.0)
Genotype 3	27 (8.1)
Genotype 4	1 (0.3)
Genotype 6	1 (0.3)
Genotype undetermined	9 (2.7)

The dominant genotype was type 1 (1a: 60.9%, 1b: 18.8%). The proportions of type 2 and type 3 were similar (9.0% versus 8.1%).

**Table 3 bioengineering-09-00509-t003:** Generalized linear equation for HCV genotypes and urine ACR.

Independent Variable	Coefficient	Standard Error	Hypothesis Test	
			Wald Chi-Square Test	*p*-Value
(Intercept)	599.567	553.9132	1.172	0.279
Genotype 1a	66.698	93.5948	0.508	0.476
Genotype 1b	−29.669	131.0363	0.051	0.821
Genotype 2	635.457	181.5780	12.247	<0.001 ***
Genotype 3	−157.011	199.4850	0.619	0.431
HCV RNA negative (ref)				
Male	−46.886	82.3127	0.324	0.569
Mexican-American	−20.579	215.4718	0.009	0.924
Hispanic	352.414	220.6777	2.550	0.110
Non-Hispanic white	69.235	190.8292	0.132	0.717
Non-Hispanic black	85.560	199.1391	0.185	0.667
Other (ref)	0 ^a^			
High school	190.048	141.5909	1.802	0.180
College or equivalent	149.012	149.3898	0.995	0.319
Post-graduate (ref)	0 ^a^			
Smoker	−188.896	85.1148	4.925	0.026 *
Diabetes mellitus	592.153	114.6121	26.694	<0.001 ***
Hepatitis B	1894.796	357.1796	28.142	<0.001 ***
Alcohol use	94.895	112.4882	0.712	0.399
Age	−11.892	6.9587	2.921	0.087
Age 20–39 years	−316.498	269.1206	1.383	0.240
Age 40–59 years	−58.615	140.2000	0.175	0.676
Age ≥ 60 years (ref)	0 ^a^			
BMI: Underweight	48.622	310.0694	0.025	0.875
BMI: Overweight	23.14	115.8813	0.04	0.842
BMI	−6.205	7.9142	0.615	0.433
BMI: Normal (ref)	0 ^a^			

* *p*-Value < 0.05, *** *p*-Value < 0.001. ^a^: Reference. Controlled for sex, race, education level, smoking, diabetes, hepatitis B, alcohol use, and BMI. Abbreviations: BMI: body mass index; HCV: hepatitis C virus.

**Table 4 bioengineering-09-00509-t004:** Several mechanisms of HCV-associated kidney diseases.

Categories	Cryoglobulin Type	Phenotype of GN	Histological Findings
Cryoglobulinemic GN			
	Type I: isolated monoclonal IgA, IgM, or IgG Type II: IgG and a monoclonal IgM RF Type III: IgG and a polyclonal IgM RF	Membranoproliferative GN (frequently associated with type II cryoglobulinemia)	IC deposition in: - the lumen of glomerular capillaries (eosinophilic thrombi) - the subendothelium of capillary walls and endothelitis by the complement activation - the mesangium, caused by the high affinity for fibronectin in the mesangial matrix - impaired clearance of ICs by macrophages and monocytes
Non-cryoglobulinemic GN			
		Membranoproliferative GN	Mesangial deposition of IC with viral-like particles, IgG and other complement fractions
		Membranous GN	IC containing HCV proteins deposition in subepithelial glomerular basement membrane
		IgA nephropathy	Impaired IgA clearance and IgA-containing IC
		FSGS	Possible HCV direct injury to the podocytes
		Fibrillary and immunotactoid glomerulopathy	Extracellular deposition of microfibrils within the mesangium and glomerular capillary walls (predominance of IgG4 deposition)

Abbreviations: HCV: hepatitis C virus; FSGS: focal segmental glomerulosclerosis; GN: glomerulonephritis; IC: immunocomplexes; Ig: immunoglobulin; RF: rheumatoid factor.

## Data Availability

The data underlying this article will be shared upon reasonable request to the corresponding author.

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
