# Peer review of "From Bench to Bedside: Clinical and Biomedical Investigations on Hepatitis C Virus (HCV) Genotypes and Risk Factors for Albuminuria"

_bioengineering, 2022, doi:10.3390/bioengineering9100509_

Round 1

Reviewer 1 Report

In the current work, the authors investigated kidney function in patients with and without HCV infections. They report HCV genotype 2 as a risk factor for albuminuria in comparison with other HCV genotypes. The study is interesting but this topic has been previously discussed. Also the number of HCV infected patients, especially those with genotype 2, is relatively low.

Author Response

Response to Reviewer 1

[General Comment]

In the current work, the authors investigated kidney function in patients with and without HCV infections. They report HCV genotype 2 as a risk factor for albuminuria in comparison with other HCV genotypes. The study is interesting but this topic has been previously discussed. Also the number of HCV infected patients, especially those with genotype 2, is relatively low.

Author Reply: We sincerely appreciate your time and effort spent reviewing this manuscript. We have revised the manuscript thoroughly according to your suggestions. The responses to your comments are found below. Please see the section of Introduction.

The Third United States (US) National Health and Nutrition Examination Survey (NHANES) showed that HCV infection is related to albuminuria, especially in populations aged 60 years or older [9]. Taiwanese researchers showed in 1991-1992 that patients with chronic HCV infection are more likely to have higher proteinuria than those without chronic HCV infection. Moreover, a high level of HCV- ribonucleic acid (RNA) (viral load) and genotype 2 have been reported to be a strong predictor of CKD [10]. Contrarily, the REVEAL-HCV study recommended that HCV genotype 1 and high serum HCV-RNA levels could be strong predictors of end-stage renal disease (ESRD) [11]. In 2018, a study using data from the National Health and Nutrition Database and the National Vital Statistics System (NVSS) showed that sex and race are significant factors for HCV infection. Increased urinary albumin excretion (albuminuria) is a sensitive marker for renal dysfunction, which has been demonstrated to be a risk factor for progressive kidney injury and can increase further comorbidities and mortality [12]. HCV genotype 1 has been reported to be the most prevalent globally (49.1%), followed by genotype 3 (17.9%), 4 (16.8%) and 2 (11.0%) [1]. In the US, genotype 2 accounts for approximately 13 to 15% of all HCV infections [1,2]. Different databases and types of methods commonly used by investigators were reviewed to get a global picture of biomedical techniques and computational tools used in the HCV human study. Albuminuria is a major risk factor for nephropathies and may cause the progression of CKD in patients with HCV infection. To date, there is a difference in kidney outcome between various HCV genotypes remains inconclusive. The association between different HCV genotypes and albuminuria is rarely reported, which still needs further examination among different populations of race and ethnicity [10,11]. This study aimed to investigate the clinical characteristics, HCV genotypes and risk factors for albuminuria in patients with positive HCV-RNA compared to those with negative HCV-RNA.

Last, we are deeply honored by the time and effort you spent reviewing this manuscript. In reviewing and revising our manuscript, we are motivated to read more and thus learn more from your criticisms.

Reviewer 2 Report

I have read with great interest the manuscript by Hsiao et al in which they analyze the relationships between HCV infection and various aspects derived from an extensive national registrar, the NHANES.

Minor concerns: 

1. It would be perhaps interested to account for cirrhosis or hepatic insufficiency when assessing the data.

2. Also, if available, could DAA data be analyzed in conjunction with the other variables present in the database? If not available, could the authors comment on the issue?

3. Minor spelling and punctuation corrections are needed.

Author Response

Response to Reviewer 2

[General Comment]

I have read with great interest the manuscript by Hsiao et al in which they analyze the relationships between HCV infection and various aspects derived from an extensive national registrar, the NHANES.

Author Reply: We sincerely appreciate your time and effort spent reviewing this manuscript. We have revised the manuscript thoroughly according to the reviewer’s suggestions. The responses to your comments are found below.

Minor concerns:

[Comment] 

  1. It would be perhaps interested to account for cirrhosis or hepatic insufficiency when assessing the data.

Author Reply: Thank you for your valuable comments. We have added the associated data including liver functions: AST, ALT, total bilirubin, and lipid profiles: triglyceride, cholesterol, LDL and HDL in the Text and Table 1.

3.1.13. Liver function and lipid profiles

Participates with positive HCV-RNA had significant higher liver function tests (AST, ALT) and lower triglyceride levels compared to those with negative HCV-RNA. However, there were no obvious significances between the different HCV genotypes (Table 1).

[Comment]

  1. Also, if available, could direct-acting antiviral (DAA) data be analyzed in conjunction with the other variables present in the database? If not available, could the authors comment on the issue?

Author Reply: Thank you for your valuable comments. In the present study, we collected 10-year data from five surveys: 2007-2016, with approximately 5000 samples per year. US Food and Drug Administration (FDA) approved most DAA treatment was around 2016; therefore, we did not evaluate the DAA data in this study. However, we have provided the DAA-associated information in the section of Discussion and limitation.

It has been around 5 years since the first approval use of DAA in US, and some data has been presented to have an effective control of HCV-associated kidney diseases. Most studies have shown that a stabilization of or mild rise in the eGFR rather than a return to normal kidney function [32-37]. Sise ME et al. performed a retrospective observational cohort study, which demonstrated that albuminuria improved significantly in patients without diabetes mellitus, but not in those with diabetes mellitus. Predictors of eGFR improvement included having CKD at baseline and being non-diabetic. They concluded that DAA treatment for HCV infection may slow down the progression of CKD [38]. Recently, Liu CH et al. conducted a prospective cohort study among 1987 patients with estimated glomerular filtration rate (eGFR) ≥15 mL/min/1.73m2 receiving interferon or DAA treatment. They also recommended that the kidney function and risk of ESRD were significantly improved in patients who completed sustained virologic response with anti-HCV therapy [39]. In summary, CKD has an unavoidably progressive natural course, and studies that fail to prove a favorable effect in the post-treatment period should be interpreted by com-paring them with studies that analyzed eGFR slopes before and after DAA treatment. These studies results recommend the benefits of DAA treatment in slowing down the renal function decline [37-39].

References

  1. Sise, M. E., Backman, E., Ortiz, G. A., Hundemer, G. L., Ufere, N. N., Chute, D. F., Brancale, J., Xu, D., Wisocky, J., Lin, M. V., Kim, A. Y., Thadhani, R., & Chung, R. T. (2017). Effect of Sofosbuvir-Based Hepatitis C Virus Therapy on Kidney Function in Patients with CKD. Clinical journal of the American Society of Nephrology : CJASN. 12(10), 1615–1623. https://doi.org/10.2215/CJN.02510317.
  2. Aby, E. S., Dong, T. S., Kawamoto, J., Pisegna, J. R., & Benhammou, J. N. (2017). Impact of sustained virologic response on chronic kidney disease progression in hepatitis C. World journal of hepatology, 9(36), 1352–1360. https://doi.org/10.4254/wjh.v9.i36.1352.
  3. Ogawa, E., Furusyo, N., Azuma, K., Nakamuta, M., Nomura, H., Dohmen, K., Satoh, T., Kawano, A., Koyanagi, T., Ooho, A., Takahashi, K., Kato, M., Shimoda, S., Kajiwara, E., Hayashi, J., & Kyushu University Liver Disease Study (KULDS) Group (2018). Elbasvir plus grazoprevir for patients with chronic hepatitis C genotype 1: A multicenter, real-world cohort study focusing on chronic kidney disease. Antiviral research, 159, 143–152. https://doi.org/10.1016/j.antiviral.2018.10.003.
  4. Alric, L., Ollivier-Hourmand, I., Bérard, E., Hillaire, S., Guillaume, M., Vallet-Pichard, A., Bernard-Chabert, B., Loustaud-Ratti, V., Bourlière, M., de Ledinghen, V., Fouchard-Hubert, I., Canva, V., Minello, A., Nguyen-Khac, E., Leroy, V., Saadoun, D., Trias, D., Pol, S., & Kamar, N. (2018). Grazoprevir plus elbasvir in HCV genotype-1 or -4 infected patients with stage 4/5 severe chronic kidney disease is safe and effective. Kidney international, 94(1), 206–213. https://doi.org/10.1016/j.kint.2018.02.019.
  5. Tsai, M. C., Lin, C. Y., Hung, C. H., Lu, S. N., Tung, S. Y., Chien, R. N., Lin, C. L., Wang, J. H., Chien-Hung, C., Chang, K. C., Hu, T. H., & Sheen, I. S. (2019). Evolution of renal function under direct-acting antivirals treatment for chronic hepatitis C: A real-world experience. Journal of viral hepatitis, 26(12), 1404–1412. https://doi.org/10.1111/jvh.13193.
  6. Chiu, S. M., Tsai, M. C., Lin, C. Y., Chen, C. H., Lu, S. N., Hung, C. H., Sheen, I. S., Chien, R. N., Lin, C. L., Hu, T. H., Cheng, Y. F., & Chen, C. L. (2020). Serial changes of renal function after directly acting antivirals treatment for chronic hepatitis C: A 1-year follow-up study after treatment. PloS one, 15(4), e0231102. https://doi.org/10.1371/journal.pone.0231102.
  7. Sise, M. E., Chute, D. F., Oppong, Y., Davis, M. I., Long, J. D., Silva, S. T., Rusibamayila, N., Jean-Francois, D., Raji, S., Zhao, S., Thadhani, R., & Chung, R. T. (2020). Direct-acting antiviral therapy slows kidney function decline in patients with Hepatitis C virus infection and chronic kidney disease. Kidney international, 97(1), 193–201. https://doi.org/10.1016/j.kint.2019.04.030
  8. Liu, C. H., Lin, J. W., Liu, C. J., Su, T. H., Wu, J. H., Tseng, T. C., Chen, P. J., & Kao, J. H. (2022). Long-term Evolution of Estimated Glomerular Filtration Rate in Patients With Antiviral Treatment for Hepatitis C Virus Infection. Clinical gastroenterology and hepatology : the official clinical practice journal of the American Gastroenterological Association, S1542-3565(22)00109-4. Advance online publication. https://doi.org/10.1016/j.cgh.2022.01.050.

[Comment]

  1. Minor spelling and punctuation corrections are needed.

Author Reply: Thank you for your valuable comments. We have made the corrections and sent our manuscript for English Editing again.

Last, we are deeply honored by the time and effort you spent reviewing this manuscript. In reviewing and revising our manuscript, we are motivated to read more and thus learn more from your criticisms.

Reviewer 3 Report

Dear Editor,

I have read with great interest the manuscript submitted by Hsiao et al., despite the interesting aims, the results are not well presented and/or correctly obtained: issues which need to be addressed in the revised version. 

- Is p-value two-tailed? The authors should clarify.

- In the statistical section the authors claim to perform regression analysis, however it is not clear how the model are chosen, data on calibration, etc. In fact, from what appear from the results it seems that author selected all the variables and introduced the in the SPSS. This is not how you perform multivariable regression. I suggest citing and take cue from this other MDPI article: DOI - 10.3390/diagnostics10090619

- There is a great redundancy between data presented in results between section 3.1.1 and 3.1.12, given the fact that they have already reported in TABLE 1. I suggest removing this part from results by referencing to the table.

- 3.3: What the authors  report in this section is not univariate analysis but just means and SD of various group.

Author Response

Response to Reviewer 3

[General Comment]

Dear Editor,

I have read with great interest the manuscript submitted by Hsiao et al., despite the interesting aims, the results are not well presented and/or correctly obtained: issues which need to be addressed in the revised version. 

Author Reply: We sincerely appreciate your time and effort spent reviewing this manuscript. We have revised the manuscript thoroughly according to the reviewer’s suggestions. The responses to your comments are found below.

[Comment]

- Is p-value two-tailed? The authors should clarify.

Author Reply: Thank you for your valuable comments. The p-value is two-tailed. We have added this explanation in the text.

2.3.2. Statistical analysis

Required data from the target database were selected. After the data were reviewed for completeness, IBM Statistics SPSS v24.0 was used for data classification, variable conversion, redefining variables, observation value selection, and statistical analysis, including descriptive analysis and inferential analysis, according to the specified pur-pose of the study. p < 0.05 (two-tailed) was considered statistically significant. Multi-variable generalized linear model was performed in the descriptive analysis [16], we calculated the frequency and percentage, including sex, race, HCV genotype, education level, smoking, alcohol use, diabetes mellitus, hypertension, drug use, HIV, HBV, and BMI. Furthermore, we analysed the means and standard deviation at continuous vari-ables, containing age and urine ACR. In the inferential analysis, we used Chi-square test, Kruskal-Wallis test to analyse the relationship between the HCV genotypes and urine ACR.

[Comment]

- In the statistical section the authors claim to perform regression analysis, however it is not clear how the model are chosen, data on calibration, etc. In fact, from what appear from the results it seems that author selected all the variables and introduced the in the SPSS. This is not how you perform multivariable regression. I suggest citing and take cue from this other MDPI article: DOI - 10.3390/diagnostics10090619

Author Reply: Thank you for your valuable comments. We have added this explanation in the text according to your suggestion. In addition, we added this reference in our revised manuscript. Please see the section of Materials and Methods.

Reference

  1. Sambataro, G.; Giuffrè, M.; Sambataro, D.; Palermo, A.; Vignigni, G.; Cesareo, R.; Crimi, N.; Torrisi, S.E.; Vancheri, C.; Malatino, L.; Colaci, M.; Del Papa, N.; Pignataro, F.; Roman-Pognuz, E.; Fabbiani, M.; Montagnani, F.; Cassol, C.; Cavagna, L.; Zuccaro, V.; Zerbato, V.; Maurel, C.; Luzzati, R.; Di Bella, S. The Model for Early COvid-19 Recognition (MECOR) Score: A Proof-of-Concept for a Simple and Low-Cost Tool to Recognize a Possible Viral Etiology in Community-Acquired Pneumonia Patients during COVID-19 Outbreak. Diagnostics 2020, 10, 619. https://doi.org/10.3390/diagnostics10090619.

[Comment]

- There is a great redundancy between data presented in results between section 3.1.1 and 3.1.12, given the fact that they have already reported in TABLE 1. I suggest removing this part from results by referencing to the table.

Author Reply: Thank you for your valuable comments. We made shortening this section and made the corrections.

[Comment]

- 3.3: What the authors report in this section is not univariate analysis but just means and SD of various group.

Author Reply: Thank you for your valuable comments. We have made the corrections in the text.

  • HCV Genotypes and urine ACR

To analyse the relationship between HCV genotypes and urine ACR, the values of these biochemical indicators are expressed as mean ± standard deviation. For participants with positive HCV-RNA (n = 330), the mean urine ACR was 129.3722 mg/g. Genotype 1a was identified in 200 participants (mean ACR: 124.9069 mg/g); genotype 1b was identified in 62 participants (mean ACR: 39.7754 mg/g); genotype 2 was identified in 29 participants (mean ACR: 501.6887 mg/g); and genotype 3 was identified in 27 participants (mean ACR: 13.0804 mg/g). For participants with negative HCV-RNA (n = 235), the mean urine ACR was 38.0117 mg/g.

Last, we are deeply honored by the time and effort you spent reviewing this manuscript. In reviewing and revising our manuscript, we are motivated to read more and thus learn more from your criticisms.

Reviewer 4 Report

Hsiao et al. assessed the relationship of HCV genotype on albuminuria in HCV patients. They concluded that HCV genotype 2 was associated with albuminuria with the use of NHANES cohort.

1. The major limitation of the study was the erroneous adoption of the control group, making the risk estimation not correct following analysis. The control group (no HCV) should be anti-HCV (-) patients (N = 14454). In 237 participants with HCV RNA negativity, the may include anti-HCV (+) or anti-HCV (-) patients. In patients with anti-HCV (+) but HCV RNA negative patients, they may be chronic HCV carriers who had been successfully treated with anti-HCV therapies, or those with spontaneous viral clearance after acute HCV infection. Therefore, taking the 237 participants who may have exposed to HCV infection (that is HCV viremia during some time of the course) as the reference group was not correct. In patients with anti-HCV (-), the HCV RNA positivity rate was quite low, making the false positive rate of actual HCV carriers to be extremely low, although some patients with acute HCV infection may have anti-HCV seroreversion from anti-HCV (+) to anti-HCV (-), who were expected to have very limited time frame of HCV exposure. Therefore, the author should reanalyze all the data in the current study and rewrite the findings/discussion.

2. The Figure 1 flow is not correctly depicted. Please refer to other papers to draw a correct one.

3. It was not clear about the use of regression analysis to assess the effect of HCV genotype on urine ACR. In the Method, the authors stated logistic regression analysis without mentioning the type of dependent variables (urine ACR (-) vs. urine ACR (+); or multiple dependent variables, including urine ACR (-) or other 3 different grades of urine ACR). However, in other statement, the authors used “generalized” linear regression model, which actually confused me. Please state clearly regarding the outcome grouping in the dependent variable so that it will not confuse the readers.

4. Please cite, compare, and discuss the different conclusion reached using the same NHANES cohort (Chen YC, et al. PLoS One 2022;17:e0271197)

5. Please discuss the potential effect of changes of CKD risk factors following anti-HCV treatment, which was quite relevant to the current work, with emphasis on the HCV genotype on the evolution of CKD risk factor following HCV eradication with anti-HCV agents (Sise ME, et al. Kidney Int 2020;97:193-201; Liu CH, et al. Clin Gastroenterol Hepatol 2022 Epub ahead of print).

Author Response

Response to Reviewer 4

[General Comment]

Hsiao et al. assessed the relationship of HCV genotype on albuminuria in HCV patients. They concluded that HCV genotype 2 was associated with albuminuria with the use of NHANES cohort.

Author Reply: We sincerely appreciate your time and effort spent reviewing this manuscript. We have revised the manuscript thoroughly according to the reviewer’s suggestions. The responses to your comments are found below. Please see the section of Introduction.

[Comment]

  1. The major limitation of the study was the erroneous adoption of the control group, making the risk estimation not correct following analysis. The control group (no HCV) should be anti-HCV (-) patients (N = 14454). In 237 participants with HCV RNA negativity, the may include anti-HCV (+) or anti-HCV (-) patients. In patients with anti-HCV (+) but HCV RNA negative patients, they may be chronic HCV carriers who had been successfully treated with anti-HCV therapies, or those with spontaneous viral clearance after acute HCV infection. Therefore, taking the 237 participants who may have exposed to HCV infection (that is HCV viremia during some time of the course) as the reference group was not correct. In patients with anti-HCV (-), the HCV RNA positivity rate was quite low, making the false positive rate of actual HCV carriers to be extremely low, although some patients with acute HCV infection may have anti-HCV seroreversion from anti-HCV (+) to anti-HCV (-), who were expected to have very limited time frame of HCV exposure. Therefore, the author should reanalyze all the data in the current study and rewrite the findings/discussion.

Author Reply: Thank you for your very invaluable comments. In this study, we only want to make the investigation on HCV genotypes and risk factors for albuminuria in patients with positive of HCV-RNA compared to those with negative HCV-RNA. Therefore, we used the negative HCV RNA tests for the control group. In order to avoid the confusing, we have revised the title our revised manuscript to From Bench to Bedside: Clinical and Biomedical Investigations on Hepatitis C Virus (HCV) Genotypes and Risk Factors for Albuminuria in Patients with Positive HCV-RNA. Additionally, we have added the tests associated the diagnosis for chronic and acute HCV infection in the section of limitation.

[Comment]

  1. The Figure 1 flow is not correctly depicted. Please refer to other papers to draw a correct one.

Author Reply: Thank you for your very invaluable comments. In this study, we only want to make the investigation on HCV genotypes and risk factors for albuminuria in patients with positive of HCV-RNA. Therefore, we used the negative HCV RNA tests for the control group. Inclusion criteria was participants who had results of HCV-RNA tests, and those with negative HCV-RNA tests were defined as the control group (negative HCV-RNA). The participants who had detectable positive HCV-RNA tests were defined as the positive HCV-RNA group. Participants without any detailed HCV-RNA data including HCV genotypes or lack of detailed medical information (2.1.1. to 2.1.5) were excluded in the study. The survey consisted of six items: ……..We have made a correction of Figure 1.

[Comment]

  1. It was not clear about the use of regression analysis to assess the effect of HCV genotype on urine ACR. In the Method, the authors stated logistic regression analysis without mentioning the type of dependent variables (urine ACR (-) vs. urine ACR (+); or multiple dependent variables, including urine ACR (-) or other 3 different grades of urine ACR). However, in other statement, the authors used “generalized” linear regression model, which actually confused me. Please state clearly regarding the outcome grouping in the dependent variable so that it will not confuse the readers.

Author Reply: Thank you for your very invaluable comments. In this study, we used generalized linear equation for HCV genotypes and urine ACR. We used the reference: controlled for sex, race, education level, smoking, diabetes, hepatitis B, alcohol use, and BMI. We amended the manuscript as the following as well.

3.4. Generalized Linear Equation of the Relationship between HCV Genotype and urine ACR

The results of the normality test of the linear model invalidated the hypothesis of normality. Therefore, the generalized linear model was used to analyse which HCV genotypes might predict the relationship among urine ACR. Hypertension, drug use, and HIV were excluded from analysis because of the high number of missing values and subsequent small sample sizes.

With the negative HCV-RNA group as the control group, the B value was 100.054 for genotype 1a, -44.124 for genotype 1b, 257.495 for genotype 2, and -194.592 for genotype 3. The albumin level was highest for genotype 2 (B = 257.495), although the p-values did not reach statistical significance.

For education level, after controlling for other variables, the B value was 260.312 for participants with a high school education, which was higher than that of participants with a college or equivalent education (227.930, p = 0.05). For diabetes, after controlling for other variables, the B value was 494.687 for diabetic participants, which was significantly higher than that for non-diabetic participants (p < 0.001). For hepatitis B, after controlling for other variables, the B value was 3124.922 for participants with hepatitis B, which was significantly higher than that for participants without hepatitis B (p < 0.001).

Using the generalized linear model (Table 3), the independent variables were HCV genotype, sex, age, race, education level, smoking, diabetes, hepatitis B, alcohol use, and BMI; the dependent variable was urine ACR. After controlling for sex, age, race, education level, smoking, diabetes, hepatitis B, alcohol use, and BMI, the omnibus test of the coefficient showed p < 0.001, thereby invalidating the null hypothesis and indicating that the model was significant and had predictive power. With the negative HCV-RNA group as the control group, the B value was 66.698 for genotype 1a, -29.669 for genotype 1b, 635.457 for genotype 2, and -157.011 for genotype 3. ACR was highest for genotype 2 (B = 635.457; p < 0.001). The p-values of the other three groups did not reach statistical significance. After controlling for other variables, HCV genotype 2 was significant related to urine ACR.

For diabetes mellitus, after controlling for other variables, the B value was 592.153 for diabetic participants, which was significantly higher than that for non-diabetic participants (p < 0.001). For hepatitis B, after controlling for other variables, the B value was 1894.796 for participants with hepatitis B, which was significantly higher than that for participants without hepatitis B (p < 0.001).

Table 3. Generalized linear equation for HCV genotypes and urine ACR.

[Comment]

  1. Please cite, compare, and discuss the different conclusion reached using the same NHANES cohort (Chen YC, et al. PLoS One 2022;17:e0271197)

Author Reply: Thank you for your very invaluable comments. In this study, we only want to make the investigation on HCV genotypes and risk factors for albuminuria in patients with positive HCV-RNA compared to those with negative HCV-RNA. Therefore, we used the negative HCV-RNA tests for the control group. From a recent research, Chen YC et al. performed a population-based cross-sectional study using the NHANES data from 1999-2018 to investigate the association between the status of HCV infection and risk of kidney disease. Prevalent eGFR < 60 ml/min/1.73 m2 or urinary albumin/creatinine ratio ≥ 30 mg/g was defined as kidney disease in this study. They concluded that both resolved and chronic HCV infection, particularly genotype 1, were associated with higher kidney disease risk. Compared with genotype 1, our study results show genotype 2 may have greater risk of albuminuria in patients with positive HCV-RNA. This different study results may be due to the original retrospective designs, selection bias, and unmeasured confounding factors such as including and excluding criteria. We have made this discussion in the section of Discussion.

Reference

  1. Chen, Y. C., Wang, H. W., Huang, Y. T., & Jiang, M. Y. (2022). Association of hepatitis C virus infection status and genotype with kidney disease risk: A population-based cross-sectional study. PloS one. 17(7), e0271197. https://doi.org/10.1371/journal.pone.0271197.

[Comment]

  1. Please discuss the potential effect of changes of CKD risk factors following anti-HCV treatment, which was quite relevant to the current work, with emphasis on the HCV genotype on the evolution of CKD risk factor following HCV eradication with anti-HCV agents (Sise ME, et al. Kidney Int 2020;97:193-201; Liu CH, et al. Clin Gastroenterol Hepatol 2022 Epub ahead of print).

Author Reply: Thank you for your valuable comments. In the present study, we collected 10-year data from five surveys: 2007-2016, with approximately 5000 samples per year. US Food and Drug Administration (FDA) approved most DAA was around 2016; therefore, we did not evaluate the DAA data in this study. However, we have added the DAA treatment including kidney outcomes and more references in the section of Discussion and limitation.

It has been around 5 years since the first approval use of DAA in US, and some data has been presented to have an effective control of HCV-associated kidney diseases. Most studies have shown that a stabilization of or mild rise in the eGFR rather than a return to normal kidney function [32-37]. Sise ME et al. performed a retrospective observational cohort study, which demonstrated that albuminuria improved significantly in patients without diabetes mellitus, but not in those with diabetes mellitus. Predictors of eGFR improvement included having CKD at baseline and being non-diabetic. They concluded that DAA treatment for HCV infection may slow down the progression of CKD [38]. Recently, Liu CH et al. conducted a prospective cohort study among 1987 patients with estimated glomerular filtration rate (eGFR) ≥15 mL/min/1.73m2 receiving interferon or DAA treatment. They also recommended that the kidney function and risk of ESRD were significantly improved in patients who completed sustained virologic response with anti-HCV therapy [39]. In summary, CKD has an unavoidably progressive natural course, and studies that fail to prove a favorable effect in the post-treatment period should be interpreted by com-paring them with studies that analyzed eGFR slopes before and after DAA treatment. These studies results recommend the benefits of DAA treatment in slowing down the renal function decline [37-39].

References

  1. Sise, M. E., Backman, E., Ortiz, G. A., Hundemer, G. L., Ufere, N. N., Chute, D. F., Brancale, J., Xu, D., Wisocky, J., Lin, M. V., Kim, A. Y., Thadhani, R., & Chung, R. T. (2017). Effect of Sofosbuvir-Based Hepatitis C Virus Therapy on Kidney Function in Patients with CKD. Clinical journal of the American Society of Nephrology : CJASN. 12(10), 1615–1623. https://doi.org/10.2215/CJN.02510317.
  2. Aby, E. S., Dong, T. S., Kawamoto, J., Pisegna, J. R., & Benhammou, J. N. (2017). Impact of sustained virologic response on chronic kidney disease progression in hepatitis C. World journal of hepatology, 9(36), 1352–1360. https://doi.org/10.4254/wjh.v9.i36.1352.
  3. Ogawa, E., Furusyo, N., Azuma, K., Nakamuta, M., Nomura, H., Dohmen, K., Satoh, T., Kawano, A., Koyanagi, T., Ooho, A., Takahashi, K., Kato, M., Shimoda, S., Kajiwara, E., Hayashi, J., & Kyushu University Liver Disease Study (KULDS) Group (2018). Elbasvir plus grazoprevir for patients with chronic hepatitis C genotype 1: A multicenter, real-world cohort study focusing on chronic kidney disease. Antiviral research, 159, 143–152. https://doi.org/10.1016/j.antiviral.2018.10.003.
  4. Alric, L., Ollivier-Hourmand, I., Bérard, E., Hillaire, S., Guillaume, M., Vallet-Pichard, A., Bernard-Chabert, B., Loustaud-Ratti, V., Bourlière, M., de Ledinghen, V., Fouchard-Hubert, I., Canva, V., Minello, A., Nguyen-Khac, E., Leroy, V., Saadoun, D., Trias, D., Pol, S., & Kamar, N. (2018). Grazoprevir plus elbasvir in HCV genotype-1 or -4 infected patients with stage 4/5 severe chronic kidney disease is safe and effective. Kidney international, 94(1), 206–213. https://doi.org/10.1016/j.kint.2018.02.019.
  5. Tsai, M. C., Lin, C. Y., Hung, C. H., Lu, S. N., Tung, S. Y., Chien, R. N., Lin, C. L., Wang, J. H., Chien-Hung, C., Chang, K. C., Hu, T. H., & Sheen, I. S. (2019). Evolution of renal function under direct-acting antivirals treatment for chronic hepatitis C: A real-world experience. Journal of viral hepatitis, 26(12), 1404–1412. https://doi.org/10.1111/jvh.13193.
  6. Chiu, S. M., Tsai, M. C., Lin, C. Y., Chen, C. H., Lu, S. N., Hung, C. H., Sheen, I. S., Chien, R. N., Lin, C. L., Hu, T. H., Cheng, Y. F., & Chen, C. L. (2020). Serial changes of renal function after directly acting antivirals treatment for chronic hepatitis C: A 1-year follow-up study after treatment. PloS one, 15(4), e0231102. https://doi.org/10.1371/journal.pone.0231102.
  7. Sise, M. E., Chute, D. F., Oppong, Y., Davis, M. I., Long, J. D., Silva, S. T., Rusibamayila, N., Jean-Francois, D., Raji, S., Zhao, S., Thadhani, R., & Chung, R. T. (2020). Direct-acting antiviral therapy slows kidney function decline in patients with Hepatitis C virus infection and chronic kidney disease. Kidney international, 97(1), 193–201. https://doi.org/10.1016/j.kint.2019.04.030
  8. Liu, C. H., Lin, J. W., Liu, C. J., Su, T. H., Wu, J. H., Tseng, T. C., Chen, P. J., & Kao, J. H. (2022). Long-term Evolution of Estimated Glomerular Filtration Rate in Patients With Antiviral Treatment for Hepatitis C Virus Infection. Clinical gastroenterology and hepatology : the official clinical practice journal of the American Gastroenterological Association, S1542-3565(22)00109-4. Advance online publication. https://doi.org/10.1016/j.cgh.2022.01.050.

Last, we are deeply honored by the time and effort you spent reviewing this manuscript. In reviewing and revising our manuscript, we are motivated to read more and thus learn more from your criticisms.

Reviewer 5 Report

I think the Authors presents a very interesting result, finding albumin in the urine of many patients with HCV. The strenghten this observation by showing that HCV genotype 2 is statistically more prevalent amongst HCV patients with albumin secreted in blood. I think this observation alone is sufficient to warrant publication. I would like to see data on VLDL, LDL and HDL for patients included in the study. 

The Authors mention alcohol and in the Table a large percentage are alcohol users. But alcohol use comes in many flavours and I think it should be written down in the text how many units of alcohol per week the patients drink. It also could well be that what is considered alcohol use in Taiwan is just classed as social engagement in Europe.

The same for overweight patients. A large percentage of the study group is classed as overweight. But I think it is important to write down the criteria used for this classification. It maybe that this classification is different.

between Europe and Taiwan.

If possible, include a section on HCV in kidney. Does HCV infect kidney cells or how can it be explained that HCV infection leads to secretion of albumin in urine.

Author Response

Response to Reviewer 5

[General Comment]

I think the Authors presents a very interesting result, finding albumin in the urine of many patients with HCV. The strengthen this observation by showing that HCV genotype 2 is statistically more prevalent amongst HCV patients with albumin secreted in blood. I think this observation alone is sufficient to warrant publication. I would like to see data on VLDL, LDL and HDL for patients included in the study. 

Author Reply: We sincerely appreciate your time and effort spent reviewing this manuscript. We have revised the manuscript thoroughly according to the reviewer’s suggestions. The responses to your comments are found below. Please see the section of Introduction. Thank you for your valuable comments. We have added the associated data including liver functions: AST, ALT, total bilirubin, and lipid profiles: triglycerides, cholesterol, LDL and HDL in the Text and Table 1.

3.1.13. Liver function and lipid profiles

Participates with positive HCV-RNA had significant higher liver function tests (AST, ALT) and lower triglyceride levels compared to those with negative HCV-RNA. However, there were no obvious significances between the different HCV genotypes (Table 1).

[Comment]

The Authors mention alcohol and in the Table a large percentage are alcohol users. But alcohol use comes in many flavours and I think it should be written down in the text how many units of alcohol per week the patients drink. It also could well be that what is considered alcohol use in Taiwan is just classed as social engagement in Europe.

Author Reply: Thank you for your valuable comments. We divided this into the week, month, year. We have added the associated information in Table 1.

[Comment]

The same for overweight patients. A large percentage of the study group is classed as overweight. But I think it is important to write down the criteria used for this classification. It maybe that this classification is different between Europe and Taiwan.

Author Reply: Thank you for your valuable comments. The definition is as below: BMI (Underweight: < 18.5, Normal weight: 18.5 - 24.9, Overweight: ≥ 25). We have added the associated information in the footnote of Table 1.

[Comment]

If possible, include a section on HCV in kidney. Does HCV infect kidney cells or how can it be explained that HCV infection leads to secretion of albumin in urine.

Author Reply: Thank you for your valuable comments.

4.1. HCV-associated nephropathies

Previous studies have demonstrated further evidence of the relationship between hepatitis C and progression to kidney failure [8-11]. There are several mechanisms of HCV-induced kidney damage (Table 4). The most well documented HCV related glomerulopathy was type I membranoproliferative glomerulonephritis associated with type II mixed cryoglobulinemia [7,8]. Nevertheless, only selected cryoglobulinemic patients developed glomerular injury, while the majority had nonspecific clinical presentations. The evidence from HCV RNA and associated proteins in endothelial, mesangial, and tubular cells of the kidney tissue may indicate a direct cytopathic effect of HCV invasion. HCV itself also can enter the infected cells and replicate in B lymphocytes. B cell activation during acute and chronic HCV infection, which then leads to polyclonal activation and expansion of B cells producing monoclonal IgM rheumatoid factor (RF). Monoclonal IgM that are known to display RF activity, favouring the formation of cold‐precipitable immune complexes formed by the monoclonal IgM itself, HCV, and anti-HCV polyclonal IgG antibodies which responsible for glomerulonephritis. The classical complement pathway is activated when the cold‐precipitable immune complexes are deposits in the mesangial and subendothelial space. When complement component C4 is cleaved by cold‐precipitable immune complexes, it forms two fragments, C4a and C4b. This then leads to the downstream formation of C3 and C5 convertases. C3a and c5a activates the complement cascade via C3aR and C5aR, which amplifies the inflammatory response by activating neutrophils. The interaction of membrane attack complex (MAC) with C3b and C5b enhances tissue factor (TF) expression in endothelial cells (ECs) of the kidney and can further promote proinflammatory EC activation including changes of reactive oxygen species (ROS), chemokines, and nitric oxide (NO) (Figure 2). Moreover, the hybridization signals of HCV RNA in renal biopsies were detected in tubular and capillary endothelial cells [7-9]. This evidence proposes that HCV may cause glomerular as well as tubulointerstitial injury of the kidney through cryoglobulins, the HCV antibody immune complex, or a direct cytopathic effect. Consequently, HCV infection, especially in active viral reaction, may lead to kidney injury and deterioration of kidney function. Patients with chronic HCV infection can also have an increased risk of insulin resistance, which may develop during the inflammatory process and aggravates kidney injury [26,27]. A previous human study also recommended that high HCV viral load and HCV genotype 2 could be potential predictors of CKD in Taiwan [10]. Our study results also demonstrated HCV genotype 2 was strongly correlated with urine ACR. Diabetes mellitus and hepatitis B could also be factors that might affect the degree of albuminuria in patients with HCV infections. We suggest that proteinuria or albuminuria screening test is indicated for patients with chronic HCV infection. The NHANES III showed that hepatitis C was related to albuminuria in populations aged 40 years and above. For hepatitis C, seropositivity increases the risk of proteinuria by 1.47 fold (CI: 1.12-1.94, p = 0.006) [19]. Few studies have investigated the relationship between HCV genotypes and albuminuria. From a recent research, Chen YC et al. performed a population-based cross-sectional study using the NHANES data from 1999-2018 to investigate the association between the status of HCV infection and risk of kidney disease. Prevalent eGFR < 60 ml/min/1.73 m2 or urinary albumin/creatinine ratio ≥ 30 mg/g was defined as kidney disease in this study. They concluded that genotype 1 in both resolved and chronic HCV infection was associated with higher risk of kidney disease [28]. Compared to our study, our results show that genotype 2 may have greater risk of albuminuria instead of focusing on eGFR decline in patients with HCV infection. This different study results may be due to the original retrospective designs, different end-points, selection bias, and unmeasured confounding factors such as including and excluding criteria. In our study, after adjustment the confounding factors including sex, race, education level, smoking, diabetes mellitus, hepatitis B, alcohol use, and BMI, HCV genotype 2 was confirmed to be strongly associated with ACR, which may have important clinical significance.

Table 4. Several mechanisms of HCV-associated kidney diseases.

Categories

Cryoglobulin type

Phenotype of GN

Histological findings

Cryoglobulinemic GN

Type I: isolated monoclonal IgA, IgM, or IgG

Type II: IgG and a monoclonal IgM RF

Type III: IgG and a polyclonal IgM RF

Membranoproliferative GN (frequently associated with type II cryoglobulinemia)

IC deposition in:

- the lumen of glomerular capillaries (eosinophilic thrombi)

- the subendothelium of capillary walls and endothelitis by the complements activation

- the mesangium, caused by the high affinity for fibronectin in the mesangial matrix

- impaired clearance of ICs by macrophages and monocytes

Non-cryoglobulinemic GN

Membranoproliferative GN

Mesangial deposition of IC with viral-like particles, IgG and other complements fraction

Membranous GN

IC containing HCV proteins deposition in  subepithelial glomerular basement membrane

IgA nephropathy

Impaired IgA clearance and IgA-containing IC

FSGS

Possible HCV direct injury to the podocytes

Fibrillary and immunotactoid glomerulopathy

Extracellular deposition of microfibrils within the mesangium and glomerular capillary walls (predominance of IgG4 deposition)

Abbreviations: HCV: hepatitis C virus; FSGS: focal segmental glomerulosclerosis; GN: glomerulonephritis; IC: immunocomplexes; Ig: immunoglobulin; RF: rheumatoid factor.

Figure 2. Molecular mechanisms of HCV-associated cryoglobulinemic glomerulonephritis. (A) B cell activation during acute and chronic HCV infection, which then leads to polyclonal activation and expansion of B cells producing monoclonal IgM rheumatoid factor (RF). Monoclonal IgM that are known to display RF activity, favouring the formation of cold‐precipitable immune complexes formed by the monoclonal IgM itself, HCV, and anti-HCV polyclonal IgG antibodies which responsible for glomerulonephritis. (B) The classical complement pathway is activated when the cold‐precipitable immune complexes are deposits in the mesangial and subendothelial space in the kidney. When complement component C4 is cleaved by cold‐precipitable immune complexes, it forms two fragments, C4a and C4b. This then results in the downstream formation of C3 and C5 convertases. C3a and C5a activates the complement cascade via C3aR and C5aR, which amplifies the inflammatory response by activating neutrophils. The interaction of membrane attack complex (MAC) with C3b and C5b enhances tissue factor (TF) expression in endothelial cells (ECs) of the kidney and can further promote proinflammatory EC activation.

Last, we are deeply honored by the time and effort you spent reviewing this manuscript. In reviewing and revising our manuscript, we are motivated to read more and thus learn more from your criticisms.

Reviewer 6 Report

The authors have made an interesting attempt on “From Bench to Bedside: Clinical and Biomedical Investigations on Hepatitis C Virus (HCV) Genotypes and Risk Factors for Albuminuria in Patients with Chronic HCV Infection.” The manuscript is interesting; however, the authors need to justify the scientific writing manuscript. Some of the general comments are provided below:

1.     Why do different genotypes show the variable effects of albuminuria on individuals with comorbidities including hepatitis B, diabetes mellitus, and smoking?

2.     Discussion should include more studies related to HCV genotypes and albuminuria.

3.     Please explain how HCV genotype 2 was confirmed to be strongly associated with ACR.

4.     More detailed information about the sample, such as the scale, organization structure, medical level, and geographical distribution, should be described in the Data part.

5.     Are the conclusions applicable to other countries? The significance of the paper needs to be elaborated. 

Author Response

Response to Reviewer 6

[General Comment]

The authors have made an interesting attempt on “From Bench to Bedside: Clinical and Biomedical Investigations on Hepatitis C Virus (HCV) Genotypes and Risk Factors for Albuminuria in Patients with Chronic HCV Infection.” The manuscript is interesting; however, the authors need to justify the scientific writing manuscript. Some of the general comments are provided below:

Author Reply: We sincerely appreciate your time and effort spent reviewing this manuscript. We have revised the manuscript thoroughly according to the reviewer’s suggestions. The responses to your comments are found below. 

 [Comment]

  1. Why do different genotypes show the variable effects of albuminuria on individuals with comorbidities including hepatitis B, diabetes mellitus, and smoking?

Author Reply: Thank you for your valuable comments. We have added the contents in this section. Please see the section in the Discussion.

3.3. HCV Genotypes and Urine ACR

To analyse the relationship between HCV genotypes and urine ACR, the values of these biochemical indicators are expressed as mean ± standard deviation. For participants with positive HCV-RNA (n = 330), the mean urine ACR was 129.3722 mg/g. Genotype 1a was identified in 200 participants (mean ACR: 124.9069 mg/g); genotype 1b was identified in 62 participants (mean ACR: 39.7754 mg/g); genotype 2 was identified in 29 participants (mean ACR: 501.6887 mg/g); and genotype 3 was identified in 27 participants (mean ACR: 13.0804 mg/g). For participants with negative HCV-RNA (n = 235), the mean urine ACR was 38.0117 mg/g.

3.4. Generalized Linear Equation of the Relationship between HCV Genotype and urine ACR

The results of the normality test of the linear model invalidated the hypothesis of normality. Therefore, the generalized linear model was used to analyse which HCV genotypes might predict the relationship among urine ACR. Hypertension, drug use, and HIV were excluded from analysis because of the high number of missing values and subsequent small sample sizes.

For the generalized linear model, the independent variables were HCV genotype, sex, age, race, education level, smoking, diabetes, hepatitis B, alcohol use, and BMI; the dependent variable was albumin. After controlling for sex, age, race, education level, smoking, diabetes, hepatitis B, alcohol use, and BMI, the omnibus test of the coefficient showed p < 0.001 (Table 3), thereby invalidating the null hypothesis and indicating that the model was significant and had predictive power.

With the negative HCV-RNA group as the control group, the B value was 100.054 for genotype 1a, -44.124 for genotype 1b, 257.495 for genotype 2, and -194.592 for genotype 3. The albumin level was highest for genotype 2 (B = 257.495), although the p-values did not reach statistical significance.

For education level, after controlling for other variables, the B value was 260.312 for participants with a high school education, which was higher than that of participants with a college or equivalent education (227.930, p = 0.05). For diabetes, after controlling for other variables, the B value was 494.687 for diabetic participants, which was significantly higher than that for non-diabetic participants (p < 0.001). For hepatitis B, after controlling for other variables, the B value was 3124.922 for participants with hepatitis B, which was significantly higher than that for participants without hepatitis B (p < 0.001).

Using the generalized linear model, the independent variables were HCV genotype, sex, age, race, education level, smoking, diabetes, hepatitis B, alcohol use, and BMI; the dependent variable was urine ACR. After controlling for sex, age, race, education level, smoking, diabetes, hepatitis B, alcohol use, and BMI, the omnibus test of the coefficient showed p < 0.001, thereby invalidating the null hypothesis and indicating that the model was significant and had predictive power. With the negative HCV-RNA group as the control group, the B value was 66.698 for genotype 1a, -29.669 for genotype 1b, 635.457 for genotype 2, and -157.011 for genotype 3. ACR was highest for genotype 2 (B = 635.457; p < 0.001). The p-values of the other three groups did not reach statistical significance. After controlling for other variables, HCV genotype 2 was significant related to urine ACR.

For diabetes mellitus, after controlling for other variables, the B value was 592.153 for diabetic participants, which was significantly higher than that for non-diabetic participants (p < 0.001). For hepatitis B, after controlling for other variables, the B value was 1894.796 for participants with hepatitis B, which was significantly higher than that for participants without hepatitis B (p < 0.001).

 [Comment]

  1. Discussion should include more studies related to HCV genotypes and albuminuria.

Author Reply: Thank you for your valuable comments.

4.1. HCV-associated nephropathies

Previous studies have demonstrated further evidence of the relationship between hepatitis C and progression to kidney failure [8-11]. There are several mechanisms of HCV-induced kidney damage (Table 4). The most well documented HCV related glomerulopathy was type I membranoproliferative glomerulonephritis associated with type II mixed cryoglobulinemia [7,8]. Nevertheless, only selected cryoglobulinemic patients developed glomerular injury, while the majority had nonspecific clinical presentations. The evidence from HCV RNA and associated proteins in endothelial, mesangial, and tubular cells of the kidney tissue may indicate a direct cytopathic effect of HCV invasion. HCV itself also can enter the infected cells and replicate in B lymphocytes. B cell activation during acute and chronic HCV infection, which then leads to polyclonal activation and expansion of B cells producing monoclonal IgM rheumatoid factor (RF). Monoclonal IgM that are known to display RF activity, favouring the formation of cold‐precipitable immune complexes formed by the monoclonal IgM itself, HCV, and anti-HCV polyclonal IgG antibodies which responsible for glomerulonephritis. The classical complement pathway is activated when the cold‐precipitable immune complexes are deposits in the mesangial and subendothelial space. When complement component C4 is cleaved by cold‐precipitable immune complexes, it forms two fragments, C4a and C4b. This then leads to the downstream formation of C3 and C5 convertases. C3a and c5a activates the complement cascade via C3aR and C5aR, which amplifies the inflammatory response by activating neutrophils. The interaction of membrane attack complex (MAC) with C3b and C5b enhances tissue factor (TF) expression in endothelial cells (ECs) of the kidney and can further promote proinflammatory EC activation including changes of reactive oxygen species (ROS), chemokines, and nitric oxide (NO) (Figure 2). Moreover, the hybridization signals of HCV RNA in renal biopsies were detected in tubular and capillary endothelial cells [7-9]. This evidence proposes that HCV may cause glomerular as well as tubulointerstitial injury of the kidney through cryoglobulins, the HCV antibody immune complex, or a direct cytopathic effect. Consequently, HCV infection, especially in active viral reaction, may lead to kidney injury and deterioration of kidney function. Patients with chronic HCV infection can also have an increased risk of insulin resistance, which may develop during the inflammatory process and aggravates kidney injury [26,27]. A previous human study also recommended that high HCV viral load and HCV genotype 2 could be potential predictors of CKD in Taiwan [10]. Our study results also demonstrated HCV genotype 2 was strongly correlated with urine ACR. Diabetes mellitus and hepatitis B could also be factors that might affect the degree of albuminuria in patients with HCV infections. We suggest that proteinuria or albuminuria screening test is indicated for patients with chronic HCV infection. The NHANES III showed that hepatitis C was related to albuminuria in populations aged 40 years and above. For hepatitis C, seropositivity increases the risk of proteinuria by 1.47 fold (CI: 1.12-1.94, p = 0.006) [19]. Few studies have investigated the relationship between HCV genotypes and albuminuria. From a recent research, Chen YC et al. performed a population-based cross-sectional study using the NHANES data from 1999-2018 to investigate the association between the status of HCV infection and risk of kidney disease. Prevalent eGFR < 60 ml/min/1.73 m2 or urinary albumin/creatinine ratio ≥ 30 mg/g was defined as kidney disease in this study. They concluded that genotype 1 in both resolved and chronic HCV infection was associated with higher risk of kidney disease [28]. Compared to our study, our results show that genotype 2 may have greater risk of albuminuria instead of focusing on eGFR decline in patients with HCV infection. This different study results may be due to the original retrospective designs, different end-points, selection bias, and unmeasured confounding factors such as including and excluding criteria. In our study, after adjustment the confounding factors including sex, race, education level, smoking, diabetes mellitus, hepatitis B, alcohol use, and BMI, HCV genotype 2 was confirmed to be strongly associated with ACR, which may have important clinical significance.

Reference

  1. Lai, T.S.; Lee, M.H.; Yang, H.I.; You, S.L.; Lu, S.N.; Wang, L.Y.; Yuan, Y.; L’Italien, G.; Chien, K.L.; Chen, C.J. High hepatitis C viral load and genotype 2 are strong predictors of chronic kidney disease. Kidney Int. 2017, 92, 703–709; DOI:10.1016/j.kint.2017.03.021.
  2. Lai, T.S.; Lee, M.H.; Yang, H.I.; You, S.L.; Lu, S.N.; Wang, L.Y.; Yuan, Y.; L’Italien, G.; Chien, K.L.; Chen, C.J. REVEAL-HCV Study Group, Hepatitis C viral load, genotype, and increased risk of developing end-stage renal disease: reveal-HCV study. Hepatology. 2017, 66, 784-793; DOI: 10.1002/hep.29192.
  3. Chen YC, Wang HW, Huang YT, Jiang MY. Association of hepatitis C virus infection status and genotype with kidney disease risk: A population-based cross-sectional study. PLoS One. 2022 Jul 8;17(7):e0271197. doi: 10.1371/journal.pone.0271197. PMID: 35802581; PMCID: PMC9269772.

 [Comment]

  1. Please explain how HCV genotype 2 was confirmed to be strongly associated with ACR.

Author Reply: Thank you for your valuable comments.

3.4. Generalized Linear Equation of the Relationship between HCV Genotype and urine ACR

The results of the normality test of the linear model invalidated the hypothesis of normality. Therefore, the generalized linear model was used to analyse which HCV genotypes might predict the relationship among urine ACR. Hypertension, drug use, and HIV were excluded from analysis because of the high number of missing values and subsequent small sample sizes.

For the generalized linear model, the independent variables were HCV genotype, sex, age, race, education level, smoking, diabetes, hepatitis B, alcohol use, and BMI; the dependent variable was albumin. After controlling for sex, age, race, education level, smoking, diabetes, hepatitis B, alcohol use, and BMI, the omnibus test of the coefficient showed p < 0.001 (Table 3), thereby invalidating the null hypothesis and indicating that the model was significant and had predictive power.

With the negative HCV-RNA group as the control group, the B value was 100.054 for genotype 1a, -44.124 for genotype 1b, 257.495 for genotype 2, and -194.592 for genotype 3. The albumin level was highest for genotype 2 (B = 257.495), although the p-values did not reach statistical significance.

For education level, after controlling for other variables, the B value was 260.312 for participants with a high school education, which was higher than that of participants with a college or equivalent education (227.930, p = 0.05). For diabetes, after controlling for other variables, the B value was 494.687 for diabetic participants, which was significantly higher than that for non-diabetic participants (p < 0.001). For hepatitis B, after controlling for other variables, the B value was 3124.922 for participants with hepatitis B, which was significantly higher than that for participants without hepatitis B (p < 0.001).

Using the generalized linear model, the independent variables were HCV genotype, sex, age, race, education level, smoking, diabetes, hepatitis B, alcohol use, and BMI; the dependent variable was urine ACR. After controlling for sex, age, race, education level, smoking, diabetes, hepatitis B, alcohol use, and BMI, the omnibus test of the coefficient showed p < 0.001, thereby invalidating the null hypothesis and indicating that the model was significant and had predictive power. With the negative HCV-RNA group as the control group, the B value was 66.698 for genotype 1a, -29.669 for genotype 1b, 635.457 for genotype 2, and -157.011 for genotype 3. ACR was highest for genotype 2 (B = 635.457; p < 0.001). The p-values of the other three groups did not reach statistical significance. After controlling for other variables, HCV genotype 2 was significant related to urine ACR.

For diabetes mellitus, after controlling for other variables, the B value was 592.153 for diabetic participants, which was significantly higher than that for non-diabetic participants (p < 0.001). For hepatitis B, after controlling for other variables, the B value was 1894.796 for participants with hepatitis B, which was significantly higher than that for participants without hepatitis B (p < 0.001).

 [Comment]

  1. More detailed information about the sample, such as the scale, organization structure, medical level, and geographical distribution, should be described in the Data part.

Author Reply: Thank you for your valuable comments. We have made the corrections. Please see the section of Materials and Methods.

  1. Materials and Methods

2.1. Data Source and Subjects

The NHANES, a large public database, is derived from a large national survey. It is an on-going survey conducted by the Centers for Disease Control and Prevention (CDC) and the National Center for Health Statistics (NCHS) in the US [13], with the goal of generating important health statistics. The NHANES program was launched in 1960. The survey was designed for different populations and different health topics. Since 1999, it has been an on-going survey conducted every other year. It has a large sample size and includes detailed items and multiple dimensions, such as biochemistry results and a questionnaire. In general, items for older participants are more extensive. Some sensitive information may be available upon request, but most information is open to the public and can be downloaded. Participants first complete a health survey at home and then undergo a physical examination at one of four ambulatory centres. The health care team is composed of physicians, dentists, nutritionists, hygienists, and laboratory technicians, and the examinations use state-of-art high-tech equipment. The team does not provide health care per se but will provide a copy of the test results to each participant. The staff involved in the survey will also explain the results. All participant information collected during the survey is strictly confidential, and participant privacy is protected by applicable laws. All data of NHANES were collected from survey participants using the questionnaires on health-related topics in participants’ homes including physical examination and results of laboratory tests in a mobile examination center. All NHANES public data were available on the website of National Center for Health Statistics (Available from: https://www.cdc.gov/nchs/nhanes/index.htm.) and permission for publishing the analysis is not needed. The sample weights in NHANES have been constructed to adjust for non-response, oversampling, and non-coverage. Due to the thoroughness of the research methodology, NHANES data have been widely used over the years to reliably assess many diseases’ prevalence and risk factors. All participants provided written informed consent, and the NHANES protocols were approved by the research ethics review board of the National Center for Health Statistics. The NHANES sample represents the noninstitutionalized civilian U.S. population residing in all the 50 states and the District of Columbia. In addition, NHANES uses a multifarious, multistage, and probability sampling strategy instead of a simple random sample (https://wwwn.cdc.gov/nchs/nhanes/tutorials/module2.aspx).

 [Comment]

  1. Are the conclusions applicable to other countries? The significance of the paper needs to be elaborated. 

Author Reply: Thank you for your valuable comments. Different ethnic backgrounds or people in the countries may have the variation due to the human genetic or environmental effects. We have added this content in the section of limitation.

Last, we are deeply honored by the time and effort you spent reviewing this manuscript. In reviewing and revising our manuscript, we are motivated to read more and thus learn more from your criticisms.

Round 2

Reviewer 1 Report

In the revised version, the manuscript has improved according to the comments. However, the main issue regarding the numbers of HCV patients, which is relatively low, is still exist.

Author Response

Response to Reviewer 1

[General Comment]

In the revised version, the manuscript has improved according to the comments. However, the main issue regarding the numbers of HCV patients, which is relatively low, is still exist.

Author Reply: We sincerely appreciate your time and effort spent reviewing this manuscript. We have revised the manuscript thoroughly according to your suggestions. The responses to your comments are found below. Please see the section of limitation.

Genotype 1 is prevalent in the Japan, US, and Europe [1,2,30,31,40]. Therefore, the patient numbers of HCV genotype 2 were relatively low in this study.

Last, we are deeply honored by the time and effort you spent reviewing this manuscript. In reviewing and revising our manuscript, we are motivated to read more and thus learn more from your criticisms.

Reviewer 3 Report

The manuscript has been edited according to previous reviewers' reports and is ready to publication in your journal.

Author Response

Response to Reviewer 3

[General Comment]

The manuscript has been edited according to previous reviewers' reports and is ready to publication in your journal.

Author Reply: We sincerely appreciate your time and effort spent reviewing this manuscript. We have revised the manuscript thoroughly according to you and the reviewer’s suggestions.

Last, we are deeply honored by the time and effort you spent reviewing this manuscript. In reviewing and revising our manuscript, we are motivated to read more and thus learn more from your criticisms.

Reviewer 4 Report

1. Although the authors tried to modify the title by adding HCV RNA positivity in the study, it was still not accurate to exclude patients with anti-HCV (-) in the analysis. The authors have made a great misunderstanding between HCV RNA (+) and chronic HCV infection. Actually, these two terms were mutually exchangeable. If the authors tried to assess the effect of chronic HCV infection, them include all with anti-HCV (-) in the control group, as I have mentioned in the first round of review. If the authors used HCV RNA (+) in the title, then the authors should included anti-HCV (-) and HCV RNA (-) in the control group, not only the HCV RNA (-) group. This may significantly affect the results, since only 336 patients with viremia were included in the study. 

2.  It was not still clear what the method was determined for urine ACR, by dipstick testing or quantitative study? The author should state the urine ACR was determined by quantitative method . Furthermore, the authors should compare the mean or median value of urine ACR in Table 1, rather than just a ordinal parameter to mislead the readers.

3.  With regard to HCV genotyping in the flow. I found the numbers of patients to be excluded in original and revised version were not identical. Missing and indeterminate were totally different, so was the same for mixed genotype 1a/1b infection. Since the authors stated to exclude analysis in any categories of < 5 patients, the revised version of "others" combing indeterminate genotype, mixed genotypes, missing genotyping" should be put in the analysis.

4. The DAA treatment in US was launched in 2014, rather than 2018.

5. To sum up, if the authors aimed to discuss only the effect of HCV genotype on urine ACR, they should include all anti-HCV (-) patients in the control. Including only anti-HCV (+) but HCV RNA (-) may not be correct, since these patients may still exposed to HCV viremia during some period, which was eradicated later, and did not actually reflect the effects of only genotypes on urine ACR. We are still not sure the reversibility of urine ACR following successful antiviral treatment. 

Author Response

Response to Reviewer 4

[General Comment]

  1. Although the authors tried to modify the title by adding HCV RNA positivity in the study, it was still not accurate to exclude patients with anti-HCV (-) in the analysis. The authors have made a great misunderstanding between HCV RNA (+) and chronic HCV infection. Actually, these two terms were mutually exchangeable. If the authors tried to assess the effect of chronic HCV infection, they include all with anti-HCV (-) in the control group, as I have mentioned in the first round of review. If the authors used HCV RNA (+) in the title, then the authors should include anti-HCV (-) and HCV RNA (-) in the control group, not only the HCV RNA (-) group. This may significantly affect the results, since only 336 patients with viremia were included in the study.

Author Reply: We sincerely appreciate your time and effort spent reviewing this manuscript. We have revised the manuscript thoroughly according to you and the reviewer’s suggestions. Please see the section of Materials and Methods. The responses to your comments are found below.

Our study was also conducted according to the declaration of Helsinki. Inclusion criteria was participants who had the results of hepatitis C antibody (anti-HCV) or HCV-RNA tests, and those with negative HCV-RNA tests among the participates with negative anti-HCV were defined as the group of non-HCV infection. The participants who had detectable positive HCV-RNA tests were defined as the group of HCV infection. Participants without any detailed HCV-RNA data including HCV genotypes or lack of detailed medical information (2.1.1. to 2.1.5) were excluded in the study. The survey consisted of six items:

  1. It was not still clear what the method was determined for urine ACR, by dipstick testing or quantitative study? The author should state the urine ACR was determined by quantitative method. Furthermore, the authors should compare the mean or median value of urine ACR in Table 1, rather than just an ordinal parameter to mislead the readers.

Author Reply: Thank you for your valuable comments. The test of urine albumin/creatinine ratio (ACR) was determined by quantitative method. We have made this correction. Please see 2.1.4. Biochemistry.

  1. With regard to HCV genotyping in the flow. I found the numbers of patients to be excluded in original and revised version were not identical. Missing and indeterminate were totally different, so was the same for mixed genotype 1a/1b infection. Since the authors stated to exclude analysis in any categories of < 5 patients, the revised version of "others" combing indeterminate genotype, mixed genotypes, missing genotyping" should be put in the analysis.

Author Reply: Thank you for your valuable comments. We found some errors in our first version. We have made this correction. Please see Figure 1.

  1. The DAA treatment in US was launched in 2014, rather than 2018.

Author Reply: Thank you for your valuable comments. We have made this correction. Please see the section of limitation.

  1. To sum up, if the authors aimed to discuss only the effect of HCV genotype on urine ACR, they should include all anti-HCV (-) patients in the control. Including only anti-HCV (+) but HCV RNA (-) may not be correct, since these patients may still exposed to HCV viremia during some period, which was eradicated later, and did not actually reflect the effects of only genotypes on urine ACR. We are still not sure the reversibility of urine ACR following successful antiviral treatment.

Author Reply: Thank you for your valuable comments. Our study was also conducted according to the declaration of Helsinki. Inclusion criteria was participants who had the results of hepatitis C antibody (anti-HCV) or HCV-RNA tests, and those with negative HCV-RNA tests among the participates with negative anti-HCV were defined as the group of non-HCV infection. The participants who had detectable positive HCV-RNA tests were defined as the group of HCV infection. Participants without any detailed HCV-RNA data including HCV genotypes or lack of detailed medical information (2.1.1. to 2.1.5) were excluded in the study. The survey consisted of six items:

Please see the section of Materials and Methods.

Last, we are deeply honored by the time and effort you spent reviewing this manuscript. In reviewing and revising our manuscript, we are motivated to read more and thus learn more from your criticisms.

Reviewer 6 Report

The authors have addressed all the comments, so the article is acceptable for publication. 

Author Response

Response to Reviewer 6

[General Comment]

The authors have addressed all the comments, so the article is acceptable for publication. 

Author Reply: We sincerely appreciate your time and effort spent reviewing this manuscript. We have revised the manuscript thoroughly according to you and the reviewer’s suggestions.  

Last, we are deeply honored by the time and effort you spent reviewing this manuscript. In reviewing and revising our manuscript, we are motivated to read more and thus learn more from your criticisms.
